# Experimental Protocol for Phase 1 of the APARC QUOCA (QUasibiennial oscillation and Ozone Chemistry interactions in the Atmosphere) Working Group

Clara Orbe[1,2], Alison Ming[3], Gabriel Chiodo[4], Michael Prather[5], Mohamadou Diallo[6], Qi Tang[7], Andreas Chrysanthou[4], Hiroaki Naoe[8], Xin Zhou[9], Irina Thaler[10], Dillon Elsbury[11,12], Ewa Bednarz[11,12], Jonathon S. Wright[13], Aaron Match[14], Shingo Watanabe[15], James Anstey[16], Tobias Kerzenmacher[17], Stefan Versnick[17], Marion Marchand[18], Feng Li[19,20], and James Keeble[21]

[1]NASA Goddard Institute for Space Studies, New York, NY, USA
[2]Department of Applied Physics and Applied Mathematics, Columbia University, New York, NY, USA
[3]Department of Applied Mathematics and Theoretical Physics, University of Cambridge, Cambridge, UK
[4]Institute of Geosciences, Spanish National Research Council (IGEO-CSIC), Madrid, Spain
[5]University of California, Irvine, USA
[6]Institute of Climate and Energy Systems – Stratosphere (ICE-4), Forschungszentrum Jülich GmbH, 52428 Juelich, Germany
[7]Lawrence Livermore National Laboratory, Livermore, CA, USA
[8] Meteorological Research Institute, Japan
[9]School of Earth and Environment, University of Leeds, UK
[10] Department of Geosciences and Natural Resource Management, University of Copenhagen, Denmark
[11]Cooperative Institute for Research in Environmental Sciences (CIRES), University of Colorado Boulder, Boulder, CO, USA
[12]NOAA Chemical Sciences Laboratory (NOAA CSL), Boulder, CO, USA
[13]Department of Earth System Science, Tsinghua University, China
[14]Department of Earth and Atmospheric Sciences, Cornell University, Ithaca, NY, USA
[15]Japan Agency for Marine-Earth Science and Technology
[16]Canadian Centre for Climate Modelling and Analysis, Climate Research Division, Environment and Climate Change Canada, Victoria, British Columbia, Canada
[17]Institute of Meteorology and Climate Research - Atmospheric Trace Gases and Remote Sensing, Karlsruhe Institute of Technology
[18]Laboratoire Atmosphères, Milieux, Observations Spatiales, Institut Pierre-Simon Laplace, Sorbonne Université/CNRS/UVSQ, Paris, France
[19]University of Maryland Baltimore County, Baltimore, MD, USA
[20]NASA Goddard Space Flight Center, Greenbelt, MD, USA
[21]Lancaster Environment Center, Lancaster University, Lancaster, UK

**Correspondence:** Clara Orbe (clara.orbe@nasa.gov)

**Abstract.**

The quasi-biennial oscillation (QBO) is the main mode of variability in the tropical stratosphere, influencing the predictability of other regions in the atmosphere through its teleconnections to the stratospheric polar vortices and coupling to surface tropical and extratropical variability. However, climate and forecasting models consistently underestimate QBO amplitudes in

5 the lower stratosphere, likely contributing to their failure to simulate these teleconnections. One underexplored contributor to model biases is missing representation of ozone-radiative feedbacks, which enhance temperature variability in the lower stratosphere, particularly at periods at and greater than the QBO ($> 28$ months). While previous studies suggest that ozone-radiative

feedbacks can impact QBO periods, amplitudes and the associated secondary circulation in the lower stratosphere, the reported impacts differ widely among models and are hard to interpret due to differences in methodology. To this end, here we propose a coordinated experimental protocol – held joint between the Atmospheric Processes and their Role in Climate (APARC) Quasi-Biennial Oscillation Initiative (QBOi) and Chemistry Climate Modeling Initiative (CCMI) activities – which is aimed at assessing the coupling between stratospheric ozone, temperature and the circulation. We use the proposed experiments to define the ozone feedback on the QBO in both present-day and idealized (abrupt quadrupling of carbon dioxide) climates. While primary focus is on the QBO, the proposed protocol also enables analysis of other aspects of ozone-radiative-dynamical coupling in the atmosphere, including impacts on the Brewer-Dobson Circulation and tropospheric eddy-driven jet responses to future climate change. Here we document the scientific rationale and design of the QUOCA Phase 1 experiments, summarize the data request, and give a brief overview of participating models. Preliminary results using the NASA Goddard Institute for Space Studies E2-2 climate model are used to illustrate sensitivities to certain methodological choices.

## 1 Introduction

The quasi-biennial oscillation (QBO) is the main mode of variability in the tropical stratosphere, characterized by a quasi-regular period, ranging in observations between $\sim 20$ and $\sim 36$ months. The QBO also influences other regions of the atmosphere through its teleconnections to the Northern Hemisphere (NH) polar vortex (Holton and Tan (1980)) and coupling to the North Atlantic Oscillation (NAO), NH winter storm tracks (Wang et al. (2018)), subtropical jet (Garfinkel and Hartmann (2010, 2011a, b)), geopotential height anomalies over the North Pacific (Rao et al. (2020a, b)), tropical cyclones (Camargo and Sobel (2010)), and, potentially, convection associated with the Madden–Julian Oscillation (MJO) (Yoo and Son (2016)) (see Anstey et al. (2022) for a review).

In addition to its impacts on the circulation, the QBO induces significant interannual variability in the tropical and extratropical distributions of long-lived tracers such as ozone ($O_3$), methane ($CH_4$) and nitrogen oxides ($NO_y$) through its associated meridional circulation (Plumb and Bell, 1982). Meridional advection out of the tropics is typically enhanced in the upper/lower transport regimes during easterly/westerly QBO, generating variability in isopleth slopes with changing wind shear (Jones et al., 1998). At the same time, QBO easterlies inhibit isentropic mixing in the tropics, while in periods of westerly QBO winds mixing is strongly enhanced in the subtropics (O'Sullivan and Chen, 1996; Shuckburgh et al., 2001). While transport is typically thought to dominate photochemistry only below a so-called "regime transition" (located at $\sim 20$ hPa for the case of ozone), recent studies challenge the notion that transport and/or photochemistry operate in isolation in any one region, emphasizing, rather, the non-local impacts of both processes to the QBO (Ming et al., 2025).

While it is well known that the QBO influences composition, fewer studies have examined how QBO-induced variations in constituents influence the QBO itself. That is, changes in stratospheric ozone caused by QBO-driven changes in temperature and circulation alter the heating rates and overall QBO structure, resulting in a so-called "ozone feedback" (Butchart et al., 2003). Nonetheless, there is growing evidence from both models and observations that temperature variability, more generally, is enhanced in the tropical tropopause region when ozone is allowed to influence local temperatures (Yook et al. (2020)), with

ozone changes tending to amplify circulation anomalies mostly at low-frequency variability occurring with periods at (and longer than) than the QBO (Randel et al. (2021); Charlesworth et al. (2019)).

Among the most studied aspects of the coupled ozone-QBO problem is the reported ozone enhancement of QBO amplitudes in the lower stratosphere (e.g., Butchart et al. (2003); Shibata and Deushi (2005); Shibata (2021); DallaSanta et al. (2021)), where simulated QBO amplitudes are consistently too weak (Richter et al. (2020)), perhaps driving weaker-than-observed teleconnections to the NH polar vortex (Anstey et al. (2022)). Despite this finding, more recent studies have focused on other candidates to explain lower stratospheric QBO amplitude biases, including inadequate representation of Kelvin and mixed Rossby gravity waves, zonal mean wind biases, numerical diffusion, and inadequate vertical resolution (Holt et al. (2022); Simpson et al. (2025)). Perhaps this is because the analysis highlighting the influence of ozone has primarily been conducted using individual models and differing methodologies, leading to large uncertainties in both the sign and magnitude of the ozone feedback on the QBO.

In particular, whereas some studies show that interactive ozone coupling increases QBO periods by $\sim 10\%$ (Butchart et al. (2003); DallaSanta et al. (2021)) or more (Shibata and Deushi (2005)), other studies show smaller (Cordero et al. (1998)) or even negligible impacts (Cordero and Nathan (2000)). Even among the former, studies are inconsistent in whether the lengthened QBO period reflects a prolongation of the westerly (Butchart et al. (2003)) or easterly (Shibata and Deushi (2005)) phases. The influence of ozone on QBO temperatures is also uncertain, with some studies reporting a 35% peak-to-peak increase in temperature amplitudes in the middle stratosphere (Butchart et al. (2003), Figure 1b) compared to only a 2% increase in others (Shibata and Deushi (2005)). A more consistent finding is that nearly all studies report no substantial QBO wind amplitude change (Figure 1a).

Ozone can also influence the QBO secondary meridional circulation, with some studies showing a weakening of the circulation (Figure 1c), as the additional diabatic heating produced by the ozone QBO offsets the heating required to maintain thermal balance in the presence of radiative cooling (Dunkerton (1985)). That is, the downward transport of ozone elevates radiative equilibrium temperatures so that less vertical motion is needed to maintain the temperature perturbation against radiative damping (Cordero and Nathan (2000)). This mechanism, however, is not present in Shibata (2021), who find no significant changes in the residual mean upwelling associated with the QBO across simulations constrained with different ozone forcings. In addition to their impacts on advection by the residual mean circulation, waves altered by ozone may also result in enhanced (by 25/45%) westerly/easterly wave forcing of the QBO (Butchart et al., 2003).

Ozone coupling may also affect future changes in the QBO. In simulations where atmospheric carbon dioxide concentrations are abruptly quadrupled, ozone feedbacks mitigate the reduction of QBO amplitudes in the GISS E2-2 model (DallaSanta et al. (2021)) (Figure 2, top). This occurs both through a damping of the $CO_2$-induced acceleration of the Brewer-Dobson Circulation (BDC) (Hufnagl et al., 2023) and through reduced convective easterly momentum gravity wave drag deposition. However, compared to other aspects of structural uncertainty, the contribution of ozone changes to the intermodel spread of projected QBO amplitude changes remains unexplored (Richter et al. (2020)).

Given the growing, but diverse, evidence for ozone-QBO coupling (in both present-day and future climates), it is important to assess the robustness of previous claims. Methodological differences, however, preclude drawing robust conclusions from

previous single-model studies. At the same time, the relevant multi-model intercomparisons are limited either to models that lack interactive composition (QBO Initiative (QBOi); Butchart et al. (2018)) or lack an interactive QBO (Chemistry Climate Modeling Initiative (CCMI); Plummer et al. (2021)).

To this end, we introduce a new Atmospheric Processes and their Role in Climate (APARC) working group, joint between the QBOi and CCMI activities, that aims to address uncertainties in ozone-QBO coupling through a common experimental protocol leveraging the full chemistry and QBO simulation capabilities of CCMI and QBOi, respectively. The main goal of this so-called QUasibiennial oscillation and Ozone Chemistry interactions in the Atmosphere (QUOCA) group is to address the following three questions:

- **Q1**: How does ozone-temperature-dynamical coupling affect QBO periods and amplitudes in the present-day climate? How does this impact QBO teleconnections, i.e., coupling to the surface and the stratosphere-troposphere-exchange of tracers?

- **Q2**: How does ozone-temperature-dynamical coupling affect QBO periods and amplitudes in a future ($4xCO_2$) climate? How is this coupled with others changes in the large-scale circulation (e.g., Brewer Dobson Circulation and polar vortices)?

- **Q3**: Which mechanisms are associated with the ozone-temperature-dynamical coupling identified in **Q1** and **Q2**? For **Q2**, what is the relative importance of the direct stratospheric radiative and chemical changes caused by $CO_2$ (cooling) versus overall climate change (warming sea surface temperatures (SSTs))?

In Section 2 we first provide our scientific rationale for the overall framework (2.1), followed by discussion of the broader science questions that can be addressed using the protocol (2.2) and conclude with a review of the proposed experiments (2.3), with further details provided in Appendix A. Exposition of methodological choices and sensitivities are provided using the "Middle Atmosphere" NASA Goddard Institute for Space Studies (GISS) climate model (E2-2) in Section 3. An overview of participating models is then presented in Section 4 and the data request is described in Section 5, with more details presented in Appendix B. A concluding discussion is presented in Section 6.

## 2 Experimental Protocol

### 2.1 Scientific Rationale

To best inform **Q1**-**Q3**, we propose a protocol that addresses two main limitations from previous studies: inconsistent definitions of the "ozone feedback" and complexity of the coupled atmosphere-ocean response to climate change.

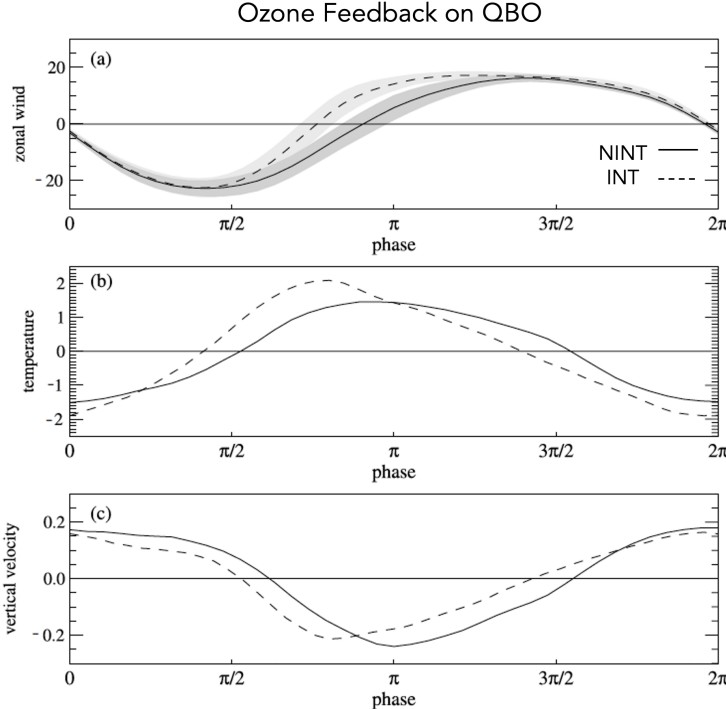

**Figure 1.** Comparisons of the tropical zonal mean zonal wind (ms$^{-1}$) (a) and anomalies in temperature (K) (b) and vertical residual velocity (mms$^{-1}$) (c) between simulations using specified (NINT, solid line) versus coupled (INT, dashed line) ozone. Fields are evaluated at the equator and at 21.5 hPa and plotted as a function of QBO phase. The data shown is from the model simulations performed using a version of the Met Office Unified Model, as reported in Butchart et al. (2003).

### 2.1.1 Inconsistent Definitions of the QBO-Ozone Feedback

The term "ozone feedback" is often used in studies to refer, more generally, to the two-way coupling between ozone and temperature (and the large-scale circulation). In practice, the feedback is almost exclusively diagnosed using models, as it is not directly extractable from observations. In simpler frameworks, one couples an analytical model with offline radiative transfer calculations to derive a "feedback parameter" (Randel et al., 2021; Ming et al., 2025), whereas when using comprehensive models, it is more common to define the feedback as the difference between so-called "interactive" (hereafter INT) and "non-interactive" (hereafter NINT) experiments run with interactive full ozone chemistry versus prescribed ozone fields, respectively.

Among the latter, studies have used various approaches to construct NINT and INT experiments. For present-day scenarios, it is common to assess the feedback by taking the difference in the QBO between a full chemistry simulation and a NINT simulation constrained with a prescribed climatological present-day ozone annual cycle, in which all induced variability except the annual cycle is removed. The latter is often taken from observations (e.g., Butchart et al. (2003); Shibata and Deushi (2005)), although this approach is not self-consistent, as the observational ozone annual cycle might deviate substantially from the

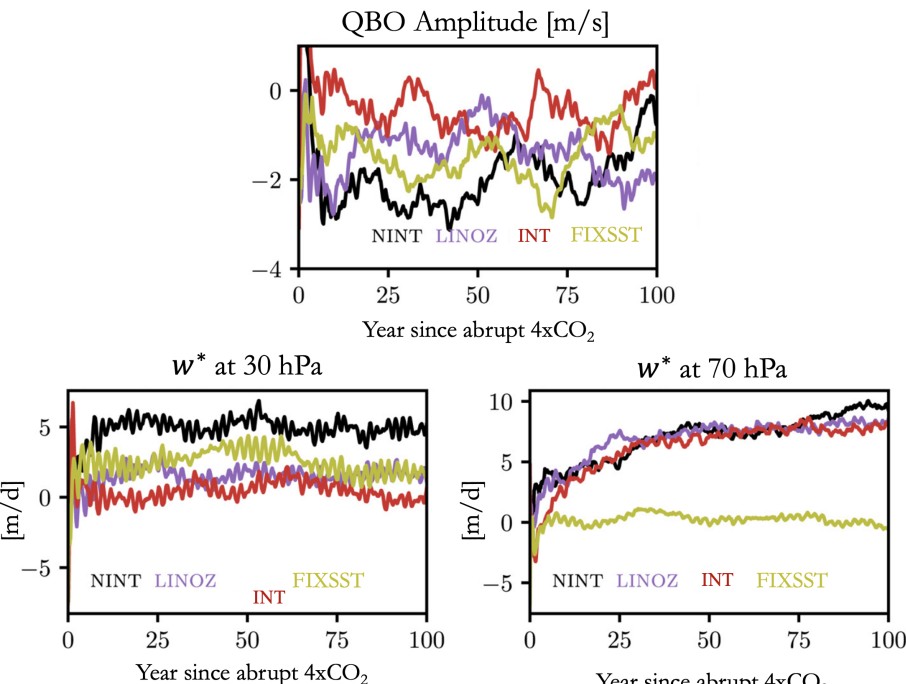

**Figure 2.** The response of the QBO amplitude (top) to an abrupt quadrupling of $CO_2$ in the CMIP6 GISS E2-2-G climate model (adapted from DallaSanta et al. (2021)). The $4xCO_2$ reduction in QBO amplitude is larger in the non-interactive simulation (NINT, black), compared to in the interactive full chemistry (OMA, red) and linearized ozone (LINOZ, purple) simulations. This is associated with a damped increase in tropical residual mean upwelling ($w^*$) at 30 hPa (bottom left), although no ozone feedback on $w^*$ is captured at 70 hPa (bottom right). Tropical averages are taken from $10°$S to $10°$N and the yellow line shows results from a non-interactive simulation constrained with preindustrial SSTs (FIXED, denoted as FT-NINT-1xCO2 in the experimental protocol outlined in Section 2.3).

ozone produced from the model's chemistry scheme. The QUOCA protocol, by comparison, promotes a more self-consistent approach in which the prescribed NINT ozone fields are derived from the model's full chemistry integrations (see Section 2.3.1 for more).

The "ozone feedback" in future climate scenarios, often refers to the degree to which ozone either amplifies or diminishes the circulation's response to $CO_2$, consistent with diagnoses of other climate feedbacks through changes in stratospheric water vapor, for example (Dessler et al., 2013), among others. The QUOCA protocol maintains this convention, noting that, in this case, the ozone feedback on the QBO includes contributions, not only from QBO-driven variability in ozone, but also from the large-scale $CO_2$-induced changes in ozone (see Section 2.3.2 for more).

## 2.1.2 Complexity of the Coupled Atmosphere-Ocean 4xCO$_2$ Response

We assert that the coupled atmosphere-ocean 4xCO$_2$ framework examined in DallaSanta et al. (2021) is too complex for addressing **Q2** (and **Q3**) robustly across models. This is because models will differ both in terms of how ozone responds to 4xCO$_2$ and how the circulation responds to that ozone perturbation. The former will hinge on a model's climate sensitivity through changes in the BDC, which are determined primarily by sea temperatures (SSTs) in the lower stratosphere (Chrysanthou et al. (2020); Abalos et al. (2021)). At the same time, the latter will depend both directly on SST changes through the BDC and through changes in the non-orographic gravity wave drag forcing of the QBO. Additional subtleties associated with running coupled atmosphere-ocean experiments – namely, use of different tunings for NINT and INT preindustrial control simulations – can introduce still more complexity. In other words, different ozone feedbacks might arise among models simply because of differences in their global surface temperature responses.

Given these limitations from previous studies, here we propose a protocol for examining QBO-ozone feedbacks focused on defining ozone feedbacks consistently across models and employing simplified AMIP configurations to minimize inter-model differences arising from climate sensitivity. While this somewhat limits immediate application of our findings to the coupled atmosphere-ocean system, we privilege the gains in understanding afforded through use of a simpler framework. Furthermore, using an AMIP framework also presents the obvious benefit that models that do not run coupled to a dynamic ocean, sea ice or land surface model are encouraged to participate.

## 2.2 Broader Science Questions

While QUOCA's main goal is to improve understanding of QBO-ozone interactions, with a focus on addressing **Q1-Q3**, these interactions occur in the broader context of other large-scale interactions with ozone (and its response to climate change). For example, DallaSanta et al. (2021) noted that the ozone feedback on the QBO period in a 4xCO$_2$ climate is linked to a so-called "damped BDC response", by which ozone feedbacks mitigate the extent to which the BDC accelerates as CO$_2$ levels are quadrupled. This feedback on background upwelling influences the QBO meridional circulation associated with the easterly phase of the QBO (eQBO), as eQBO is typically associated with enhanced upwelling in the lower tropical stratosphere (and vice versa for westerly QBO) (Figure 5 in that study). In the GISS E2-2 model, reduced eQBO amplitudes were also associated with reduced easterly momentum deposition from parameterized convective waves, further reducing the amplitude of eQBO.

Clearly, the damping of the BDC response to 4xCO$_2$ by ozone feedbacks invoked in DallaSanta et al. (2021), has broader implications beyond the QBO. In particular, consider the distinct vertical structure of the ozone feedback on the BDC featured in the GISS model, peaking in the mid-stratosphere (30 hPa; Figure 2, bottom left panel) with only minimal influence in the lower stratosphere (70 hPa; Figure 2, bottom right panel). This result highlights the distinct responses of the shallow versus deep branches of the BDC to CO$_2$ forcing (Abalos et al., 2021), although preliminary comparisons with other models suggest that the ozone feedback, particularly on the deep branch, may be model-dependent (Hufnagl et al. (2023); Calvo et al. (2025)). In addition to the BDC, the QUOCA protocol presents an opportunity to examine the stratospheric ozone feedback on the

tropospheric midlatitude eddy-driven jets in response to $4xCO_2$ forcing, including in the Northern Hemisphere, where ozone feedbacks appear to drive a negative NAO-like response (e.g., Chiodo and Polvani (2019); Li et al. (2023); Orbe et al. (2024); Wang et al. (2025)).

To this end, in addition to addressing **Q1-Q3**, the QUOCA Phase 1 experiments may also be used by the broader research community to ask:

- **Q4**: What is the ozone feedback on the $4xCO_2$ response of the stratosphere (e.g., Brewer-Dobson Circulation and polar vortices) and troposphere (e.g., eddy-driven midlatitude jets, Hadley Cell)?

Taken together, the QUOCA protocol addresses questions focused on the coupling between ozone and the QBO (**Q1-Q3**) and between ozone and other aspects of the large-scale circulation (**Q4**). Furthermore, although our AMIP framework precludes analysis of the ozone feedback on climate sensitivity, it presents a unique opportunity to examine the importance of the direct radiative response of the BDC to the large-scale circulation response to $CO_2$ forcing (Chrysanthou et al. (2020); Calvo et al. (2025)). In particular, and as detailed in the next section, the QUOCA protocol will enable decomposition of the circulation response into contributions due to changing SSTs versus local radiative effects of ozone in the stratosphere. In summary, the QUOCA protocol will afford improved understanding not only of the QBO, but, other climatological large-scale features of the atmosphere's response to future climate change.

### 2.3 Experiments

Focusing on questions **Q1-Q3** above, we propose a set of both "Present Day" (hereafter PD) experiments and "Future" (hereafter FT) experiments (Table 1). Simulations designated as "Tier 1" and "Tier 2" are required and voluntary, respectively. The Phase 1 QUOCA experiments have been designed so that:

- PD-INT minus PD-NINT represents the present-day ozone feedback on the QBO (**Q1**).

- FT-NINT-$4xCO_2$ minus PD-NINT represents the impact of $4xCO_2$ (both the direct radiative response and warmer SSTs) on the QBO with ozone fixed, while FT-INT-$4xCO_2$ minus FT-NINT-$4xCO_2$ represents the ozone feedback on the $4xCO_2$ response (**Q2**), consistent with its definition in recent studies (Chiodo et al., 2018; Chiodo and Polvani, 2019).

- FT-INT-$4xCO_2$+PDSST minus FT-INT-$4xCO_2$ and FT-INT-$1xCO_2$+4KSST minus FT-INT-$4xCO_2$ represent the QBO-ozone feedback due to global warming (+4K uniform SSTs) and stratospheric cooling ($4xCO_2$), respectively (**Q3**).

All experiments employ a time-slice framework (see 2.3.1 and 2.3.2 for further details). While a transient approach (presented in Appendix E) may be preferred, we discourage it, since analysis of transient experiments may be complicated by non-stationary trends, internal variability intrinsic to atmosphere-ocean coupling (i.e., ENSO) and anomalous triggers (e.g., volcanoes, wildfires) in ozone. Finally, for all experiments except PD-INT 3 ensemble members are requested (Col. 2, Table 1).

| Experiment | Simulation Length [years] (x Ens. Mem.) | $O_3$ | $CO_2$ | Other Trace Gases (ODS, $CH_4$, $N_2O$, tropospheric pollutants) | SSTs | SICs | Tier |
|---|---|---|---|---|---|---|---|
| *PD-INT* | 90(x1) | Interactive | CMIP6 (2000-2020)* mean | CMIP6 (2000-2020) mean | HadISST1 (2000-2020) mean | HadISST1 (2000-2020) mean | 1 |
| *PD-NINT* | 30(x3) | Climatological PD-INT** | CMIP6 (2000-2020) mean | CMIP6 (2000-2020) mean | HadISST1 (2000-2020) mean | HadISST1 (2000-2020) mean | 1 |
| *FT-NINT-4xCO₂* | 30(x3) | PD-NINT Climatology | 4xCMIP6 (2000-2020) mean | CMIP6 (2000-2020) mean | HadISST1 (2000-2020) mean + uniform 4K | HadISST1 (2000-2020) mean | 1 |
| *FT-INT-4xCO₂* | 30(x3) | Interactive | 4xCMIP6 (2000-2020) mean | CMIP6 (2000-2020) mean | HadISST1 (2000-2020) mean + uniform 4K | HadISST1 (2000-2020) mean | 1 |
| *FT-INT-4xCO₂ +PDSST* | 30(x3) | Interactive | 4xCMIP6 (2000-2020) mean | CMIP6 (2000-2020) mean | HadISST1 (2000-2020) mean | HadISST1 (2000-2020) mean | 2 |
| *FT-INT-1xCO₂ +4KSST* | 30(x3) | Interactive | CMIP6 (2000-2020) mean | CMIP6 (2000-2020) mean | HadISST1 (2000-2020) mean + uniform 4K | HadISST1 (2000-2020) mean | 2 |

**Table 1.** The proposed list of "Present-Day" (PD) and "Future" (FT) experiments. 3 ensemble members per experiment are requested, with the exception of PD-INT. *All uses of (2000-2020 mean) include the annual cycle for SSTs and SICs (obtained from input4MIPs), but just a single annual mean value for $CO_2$ and other long-lived trace gases. For tropospheric chemistry (if included), emissions of short-lived pollutants including aerosols should use a monthly mean (or equivalent) annual cycle that repeats.**Each PD-NINT ensemble is constrained with the climatological mean annual cycle of ozone, derived from each non-overlapping 30-year-long segment of the PD-INT integration.

### 2.3.1 Present-Day (PD) AMIP Simulations

1. *PD "Interactive" Experiment (PD-INT, Tier 1)*: The PD-INT experiment is a time-slice "present-day" 90-year-long integration run using full interactive chemistry and forced with boundary conditions used in CMIP6 available from

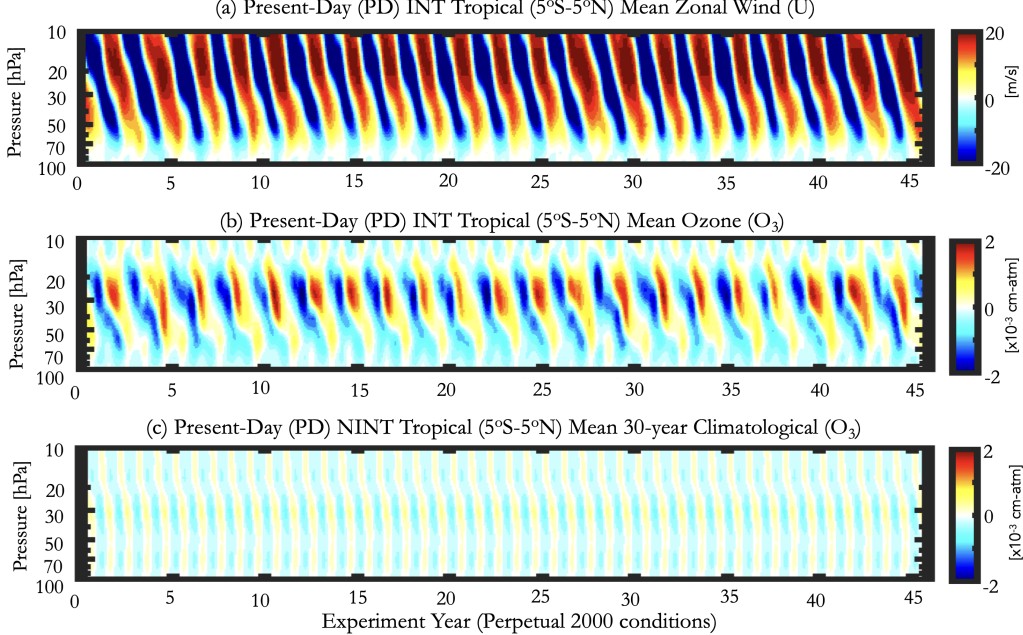

**Figure 3.** (a) The wind QBO (zonal winds, averaged over $5°$S-$5°$N in a 50-year-long GISS E2-2 time-slice INT integration and (b) the corresponding ozone QBO (ozone, also averaged over $5°$S-$5°$N). (c) The annually repeating climatological mean ozone concentrations used to force the GISS E2-2 time-slice NINT integration, derived from the first 30 years' worth of ozone from the GISS INT integration.

the input4MIPs website[1] (see Table 1). Specifically, year 2000-2020 mean values are prescribed, with the monthly seasonal cycles retained for SSTs and sea ice concentrations (SICs). A single annual mean value should be used for the long-lived gases (CO, $CH_4$, $N_2O$, ozone depleting gases) as well as solar, volcanic and (if used) biomass burning emissions, although the annual cycle in tropospheric emissions of short-lived species over 2000-2020 should be used. A 10-year spin up for chemistry is recommended and it is imperative that all participating models run with at least interactive stratospheric ozone (see Appendix D for more). For models using more simplified chemical mechanisms, at least ozone needs to run interactively coupled with the model's internal dynamics and radiation, no matter how simplified its treatment. Note that some simplified mechanisms may not capture the influence of NOx variations on ozone above $\sim$ 20 hPa, nor at lower levels, since the overlying ozone column can modulate the ultraviolet radiation that reaches the lower stratosphere and affect infrared transfer between layers (Ming et al., 2025). Analysis of the ozone feedbacks represented using such simplified schemes will therefore be one of the focal areas of the QUOCA proposed working groups (Section 6).

2. *PD "Non-interactive" Experiment (PD-NINT, Tier 1):* The PD-NINT experiment is identical to PD-INT, except that a single, annual cycle of three-dimensional, monthly-mean ozone fields are prescribed, based on a 30-year climatology

---

[1] https://esgf-node.llnl.gov/search/input4mips/

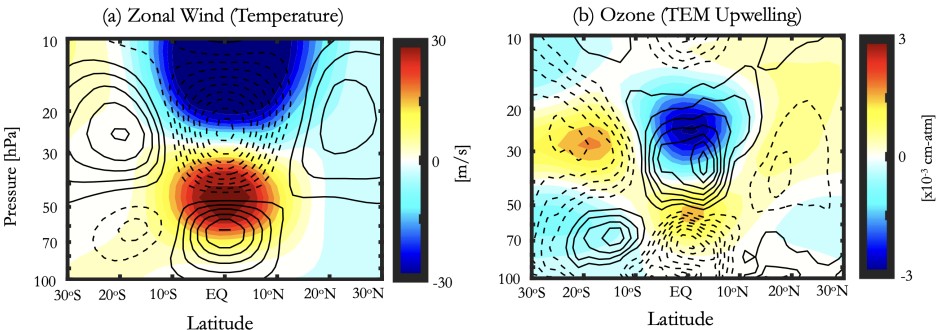

**Figure 4.** QBO westerly minus QBO easterly composites in zonal wind (colors) and temperature (black contours, 0.5 K interval) (a) derived from the GISS E2-2 PD-INT time-slice experiment. (b) Same as in (a), except for ozone (colors) and residual mean upwelling (black contours, $2*(10^{-7})$ mb/s). The QBO has been evaluated at 50 hPa.

derived from the PD-INT experiment, keeping all other compositional and boundary forcings identical. The 90-year-long PD-INT experiment should generate three successive 30-year ozone climatologies, which are used to constrain three 30-year-long PD-NINT ensemble members. Preliminary results using the GISS E2-2 model show that this method of ensemble generation produces an intra-ensemble variance in QBO response that is similar in magnitude to the spread generated using more standard approaches (i.e., random perturbations to tropospheric temperatures, as employed in the

CMIP6 AMIP E2-2 DECK submission (Rind et al., 2020)). Initialization is up to the modeling group. The difference PD-INT minus PD-NINT is used to quantify the present-day ozone feedbacks on the QBO. Note that this approach is distinct from previous studies in which an ozone climatological seasonal cycle is prescribed from an observational dataset (Butchart et al. (2003); Shibata and Deushi (2005)) and is thus certain to be different from the underlying model (in this case, PD-INT).

### 2.3.2 Future (FT) 4xCO$_2$ AMIP Simulations

1. *FT 4xCO$_2$ "Non-Interactive" Experiment (FT-NINT-4xCO$_2$, Tier 1):* The FT-NINT-4xCO$_2$ experiment is based on a climate change scenario and builds off the PD-NINT experiment in two main ways: 1) CO$_2$ concentrations are quadrupled from the values prescribed in PD-NINT and 2) a spatially uniform perturbation of +4K is applied to all SST grid points. SIC values are kept fixed to the values used in PD-NINT. Three 30-year-long ensemble members are initialized from the

three PD-NINT ensemble members and constrained with their corresponding three-dimensional ozone fields (all other chemical species are the same as in PD-NINT). See Appendix C for a note on the use of prescribed PD-NINT ozone values in this experiment.

2. *FT 4xCO$_2$ "Interactive" Experiment (FT-INT-4xCO$_2$, Tier 1):* The FT 4xCO$_2$ "Interactive" experiment is identical to FT-NINT-4xCO$_2$, except that ozone is allowed to respond interactively to the quadrupled CO$_2$ concentrations and +4K SST perturbations. Each ensemble member is initialized from a different point in the PD-INT experiment in order to sample different initial states in the atmosphere. The difference FT-INT-4xCO$_2$ minus FT-NINT-4xCO$_2$ is used to quantify 4xCO$_2$ ozone feedbacks on the QBO. Note that for whole atmospheric chemistry models running with comprehensive tropospheric chemistry, care will need to be taken in diagnosing the influence of the future climate change on tropospheric chemistry and its subsequent impacts on climate forcing and tropopause stability (primarily through ozone and aerosols). Changes in upper tropospheric and stratospheric water vapor associated with warmer SSTs will also represent a confounding factor influencing both temperatures and ozone chemistry in the stratosphere. If modeling centers can take measures to avoid these complications we advise that they do so.

3. *FT 4xCO$_2$ "Interactive" Present-Day SST Experiment (FT-INT-4xCO$_2$+PDSST, Tier 2):* The FT 4xCO$_2$ "Interactive" Fixed SST experiment is identical to FT-INT-4xCO$_2$, except that SSTs are fixed to the present-day values used in PD-INT.

4. *FT 1xCO$_2$ "Interactive" +4K SST Experiment (FT-INT-1xCO$_2$+4KSST, Tier 2):* The FT 1xCO$_2$ "Interactive" +4K SST experiment is identical to FT-INT-4xCO$_2$, except that CO$_2$ concentrations are identical to those used in PD-INT.

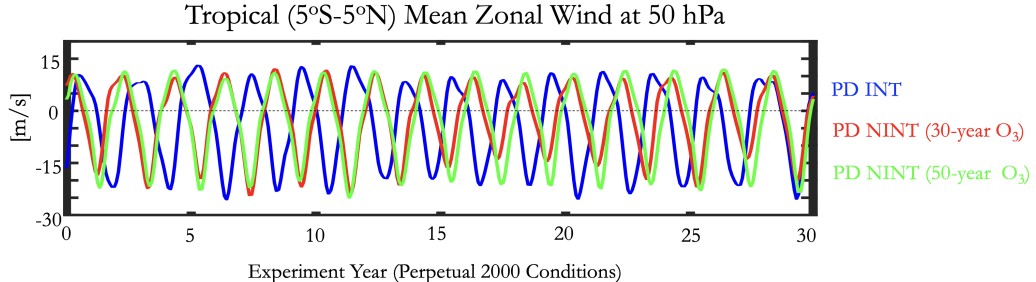

**Figure 5.** Tropical zonal winds at 50 hPa, compared between GISS E2-2 PD-INT (blue) and PD-NINT simulations using 30-year (red) versus 50-year (green) ozone climatological forcings. In the GISS E2-2 model, interactive chemistry (blue), results in a longer QBO period. Similar results apply to 30 hPa (not shown).

## 3    Exposition of Methodology using the High-Top GISS Climate Model E2-2

The GISS E2-2 model (Rind et al. (2020); Orbe et al. (2020a)) is now used to illustrate anticipated sensitivities of the derived
ozone feedback to certain methodological choices. In particular, we explore consequences of using a time-slice versus transient
framework and in making certain assumptions used to derive the NINT climatological ozone forcing.

To begin, the wind QBO is shown for a 50-year-long time-slice PD-INT experiment performed using E2-2 (Fig. 3a). The
QBO generated in this model has been validated in previous studies and shown, for AMIP configurations like that employed
here, to feature a period of 28–29 months, in agreement with observations. The simulated QBO amplitude at 30 hPa is $\sim 15\%$
too weak compared to observations (Rind et al., 2020), consistent with other CMIP6 models (Orbe et al., 2020b; Rao et al.,
2020a).

This QBO-driven wind variability affects simulated ozone (Fig. 3b), as positive ozone anomalies propagate downward during
the westerly shear phase of the QBO below $\sim 20$ hPa. This ozone QBO, generated using a linearized ozone scheme, compares
well with that simulated in model configurations using the more comprehensive interactive trace-gas and aerosol scheme used
in the CMIP6 submission of the model (OMA, Bauer et al. (2020), DallaSanta et al. (2021), Orbe et al. (2020a)). More precisely,
during the westerly phase of the QBO, anomalous downwelling associated with warmer anomalies in the tropics draws larger
values of ozone into the lower stratosphere; conversely, upwelling anomalies associated with easterly wind shear bring air
from the troposphere into the lower stratosphere, thereby reducing ozone. While below 20 hPa, the ozone variations are more
clearly modulated by transport associated with QBO dynamics, above 20 hPa, the ozone anomalies display a more complicated
relationship with the circulation, as variability in tropical ozone is controlled by variations in photochemistry.

The overall QBO amplitude, displayed as QBOW-QBOE anomalies in zonal winds, temperature and ozone (Figure 4),
features strong cooling overlying warming and anomalous negative and positive ozone anomalies associated with the QBO
meridional circulation (Plumb and Bell (1982)). Among other aspects of the ozone QBO, its associated meridional width and
the strength of background ozone gradients is expected to influence the ozone feedback on the QBO.

In addition to the PD-INT experiment, we also performed a 50-year-long PD-NINT experiment constrained using a 30-year climatological annual cycle of ozone derived from the PD-INT experiment (Fig. 3c). In the GISS model the use of prescribed ozone, in which QBO-ozone variability has been removed, results in a shorter QBO period at 50 hPa by $\sim 3 \pm 2$ months and reduced QBO amplitudes by $\sim 2 \pm 3$m/s (or 10%) (Figure 5, blue versus red lines). A similar response occurs at both 30 hPa and 40 hPa (not shown). Note that the use of a 50-year-long ozone climatology, versus 30 years, does not significantly impact

the ozone feedback (Figure 5, red versus green lines).

Experiments performed using a transient, as opposed to time-slice, framework yield very similar results (Appendix E, Figures E1-E3). In both cases, the QBO period is reduced by $\sim$ 2-3 months and the QBO amplitudes weaken, when ozone is allowed to couple interactively with the QBO. Since there is less internal variability in the QBO response using the time-slice framework, however, we recommend using this approach. For more details on the transient historical experimental setup see Appendix E.

## 4   Participating Models

Recent attempts to isolate the ozone feedback in a multi-model context have involved comparing CMIP6 models including interactive stratospheric ozone with those running with prescribed ozone (Wang et al. (2025)) by, e.g., following the approach of Morgenstern et al. (2022) of taking differences between pairs of models. While these analyses can be instructive, they do not cleanly distinguish between intermodel differences arising due to differences in ozone versus other contributors to model

structural uncertainty. That is, clean pairs of "non-interactive" and "interactive" simulations performed using the same model do not exist broadly across the CMIP6 archive or in past phases of the CCMI and QBOi projects.

To this end, we have identified 8 models that contributed either to QBOi or CCMI (or both), all of which simulate an interactive QBO and interactive ozone chemistry. These models are listed in Table 2, along with the APARC activity to which they contributed results, their respective institutes and investigators using the models and associated contact information. Model

features and associated references most relevant to simulating the QBO and ozone chemistry are shown in Table 3. For the models contributing to QBOi we refer the reader to more details presented in Butchart et al. (2018). Likewise, for models contributing to CCMI we refer the reader to Morgenstern et al. (2017) and Plummer et al. (2021).

Note that some models use linearized ozone chemistry; among these, some modeling groups plan to include submissions using both linearized and full chemistry mechanisms, which will enable assessment of the extent to which linearized param-

eterizations capture the complete dynamics of ozone-circulation coupling represented in the full chemistry simulations. We expect, however, that this will only apply to a few models, so care will be taken in generalizing any findings.

## 5   Data Request

The data request was initially based off the request for Phase 1 of QBOi (Butchart et al., 2018), modified to include a few more dynamical and thermodynamical variables from the Dynamics and Variability Model Intercomparison Project (DynVarMIP) for

CMIP6 (Gerber and Manzini (2016)), as well as aspects of the vertical grid used in the Stratospheric Nudging And Predictable

| Model | APARC Activity | Expts.[*] | Institute(s) | Investigators | Email Address |
|---|---|---|---|---|---|
| GISS E2-2-G | QBOi | 1-6 | NASA GISS[1] | Clara Orbe | clara.orbe@nasa.gov |
| GEOSCCM | CCMI | 1-4 | NASA GSFC[2] | Feng Li; | feng.li-1@nasa.gov; |
| | | | | Qing Liang | qing.liang@nasa.gov |
| CESM2 (WACCM6) | QBOi and CCMI | 1-6 | NCAR[3] | Rolando Garcia; | rgarcia@ucar.edu; |
| | | | | Gabriel Chiodo; | gabriel.chiodo@csic.es; |
| | | | | Andreas Chrysanthou | anchrysa@ucm.es |
| E3SMv3 | QBOi and CCMI | 1-6 | DOE LLNL[4] | Qi Tang; | tang30@llnl.gov; |
| | | | | Jinbo Xie | jinbo.xie@princeton.edu |
| ICON | −[**] | 1-6 | KIT[5] | Stefan Versick; | stefan.versick@kit.edu; |
| | | | | Tobias Kerzenmacher | tobias.kerzenmacher@kit.edu |
| UKESM1-StratTrop | CCMI | 1-6 | Met Office-NCAS[6] | James Keeble | j.keeble2@lancaster.ac.uk; |
| | | | | N. Luke Abraham | nla27@cam.ac.uk |
| LMDZ-Reprobus | CCMI | 1-2 | LATMOS[7], LMD[8] | Marion Marchand; | marion.marchand@latmos.ipsl.fr; |
| | | | | Francois Lott; | francois.lott@lmd.ipsl.fr; |
| | | | | David Cugnet; | david.cugnet@latmos.ipsl.fr; |
| | | | | Slimane Bekki; | slimane.bekki@latmos.ipsl.fr; |
| | | | | Lola Falletti | lola.falletti@latmos.ipsl.fr |
| AGCM3 CMAM | QBOi and CCMI | 1-2 | CCCma[9] | James Anstey; | James.Anstey@ec.gc.ca; |
| | | | | David Plummer; | David.Plummer@ec.gc.ca; |
| | | | | Barbara Winter | Barbara.Winter@ec.gc.ca |
| MIROC-ES2H | QBOi and CCMI | 1-4 | JAMSTEC[10] | Shingo Watanabe | wnabe@jamstec.go.jp |

**Table 2.** Participating models, previous APARC involvement, experiments planned, institute, lead investigators and contact information. [*]Note that experiments 1-6 refer to PD-INT, PD-NINT, FT-INT-4xCO$_2$, FT-NINT-4xCO$_2$, FT-INT-4xCO$_2$+PDSST and FT-INT-1xCO$_2$+4KSST, respectively. [**] ICON did not participate in either the QBOi or CCMI previous phases. [1]: National Aeronautics and Space Administration Goddard Institute for Space Studies; [2]: National Aeronautics and Space Administration Goddard Space Flight Center; [3]: National Center for Atmospheric Research; [4]: Department of Energy Lawrence Livermore National Laboratory; [5]: Karlsruher Institut für Technologie; [6]: Met Office, Exeter, UK; NCAS-Climate, University of Cambridge, CB2 1EW, UK [7]: Laboratoire Atmosphères, Observations Spatiales; [8]: Laboratoire de Météorologie Dynamique; [9]: Canadian Centre for Climate Modelling and Analysis (CCCma), Climate Research Division, Environment and Climate Change Canada; [10]: Japan Agency for Marine-Earth Science and Technology

Surface Impacts (SNAPSI) activity (Hitchcock et al. (2022)) in the upper troposphere/lower stratosphere region. More substantially, the QUOCA data request also includes a significant number of compositional outputs requested for CCMI (Plummer et al. (2021)). The full set of output variables is tabulated in Appendix B (Tables E1–E5), noting that the same set of variables

| Model | Horizontal Resolution (Atm) | Number of Vertical Levels | Model Lid | Gravity Wave Drag (Reference(s)) | Ozone Mechanism (Reference(s)) |
|---|---|---|---|---|---|
| GISS E2-2-G | 2° lat x 2.5° lon | 102 | $2 \times 10^{-3}$ hPa | Rind et al. (2014, 2020) | Bauer et al. (2020); McLinden et al. (2000) |
| GEOSCCM | 1° lat x 1° lon | 72 | $1 \times 10^{-2}$ hPa | McFarlane (1987); Garcia and Boville (1994) | Nielsen et al. (2017) |
| CESM2 (WACCM6) | 0.9° lat x 1.25° lon | 110 | $6 \times 10^{-6}$ hPa | Richter et al. (2010); Mills et al. (2017) | Gettelman et al. (2019) |
| E3SMv3 | 1° lat x 1° lon | 80 | $10^{-1}$ hPa | Richter et al. (2010) | Tang et al. (2025) |
| ICON | R2B5 ($\sim$ 80 km) | 150 | 78 km | Orr et al. (2010) Lott and Miller (1997) | McLinden et al. (2000) |
| UKESM1-StratTrop | 1.25° lat x 1.875° lon | 85 | 85 km | Walters et al. (2014) | Archibald et al. (2020) |
| LMDZ-Reprodbus | 2.5° lat x 1.3° lon | 79 | $10^{-2}$ hPa | Lott and Guez (2013); De la Cámara and Lott (2015); De La Camara et al. (2016) | Marchand et al. (2012) |
| AGCM3 CMAM | T47 | 80 | 95 km | Scinocca and McFarlane (2000); Scinocca (2003) | De Grandpré et al. (2000); Jonsson et al. (2004) |
| MIROC-ES2H | T85 | 90 | $4 \times 10^{-3}$ hPa | Hines (1997); McFarlane (1987); Watanabe (2008) | Watanabe et al. (2011); Kawamiya et al. (2020) |

**Table 3.** Model features related to representation of ozone coupling with the QBO, including horizontal resolution, number of vertical levels, model lid (hPa) and references for gravity wave drag and ozone chemistry schemes.

is requested from all experiments. To reduce data volume, however, some high-volume variables are requested from only one ensemble member, as described below.

## 5.1 Diagnostics

Here we provide a general description of the data request, first reviewing the common aspects shared between the QBOi Phase 1 and QUOCA data requests, which include diagnostics related to climate and variability (Table E1), dynamics (Table E2), and equatorial wave spectra (Table E5).

Table E1 is identical to Table 1 in Butchart et al. (2018), except for a few additional requests (e.g., outgoing longwave radiation, tropopause air pressure and tropopause air temperature). Table E2 is also largely based off the QBOi Phase 1 request and contains various stratospheric dynamical diagnostics requested from CMIP6 simulations as part of DynVarMIP, including the Transformed Eulerian Mean (TEM) quantities which enable analysis of the QBO zonal momentum budget and, more generally, the mean meridional circulation in the stratosphere.

Equatorial wave spectra calculated from 6-hour resolution (instantaneous, not time averaged) winds and temperature (Table E5) will enable analysis of the QBO wave driving in a consistent manner across models. This information will be particularly useful to the QUOCA project as one goal will be to understand how ozone feedbacks modify wave forcings of the QBO, as suggested in previous studies (Echols and Nathan (1996); Butchart et al. (2003)). Note that this data is requested for a specific subset of latitudes (15°S to 15°N), as in QBOi Phase 1, and for one ensemble member.

In addition to the requests derived from QBOi Phase 1, we include two additional tables (Tables E3 and E4). First, Table E3 enables, through its request of various zonal mean 6-hourly quantities, offline calculation of the TEM circulation and related Eliassen-Palm fluxes based on the recommendation in Ming (2016). This is included to account for potential inconsistencies that may arise due to different TEM formulations among the modeling centers, which may be reflected in the TEM output contributed for Table E2. Note that the 6-hourly output request in Table E3 is distinct from the limited-domain (i.e., 15°S-

15°N), albeit zonally varying, quantities included in the equatorial wave spectra request (Table E5).

      Finally, the QUOCA data request departs most from that in QBOi Phase 1 in terms of its compositional outputs, as shown in Table E4. Monthly mean (and daily, for $O_3$ and $H_2O$) distributions of a variety of trace gases relevant to stratospheric ozone chemistry, in addition to explicit loss terms, are requested. In addition, a subset of the idealized tracers included in Phase 1 of CCMI and reported in Orbe et al. (2018) are included, focusing on those most relevant to stratospheric transport and

stratosphere-troposphere-exchange (e.g., an age-of-air tracer (AOA), stratospheric ozone tracer ($O_3S$), etc.).

### 5.2   Spatial Resolution

The horizontal grid should be a latitude-longitude grid equivalent to the original model resolution. In terms of vertical resolution, output is requested for all variables in Tables B1-B2 and Table B4 on a standard 42 pressure level grid (hereafter "plev42") that reflects a slight modification from the 39-pressure level grid used in DynVarMIP to be more in line with what was used in

SNAPSI, providing more levels in the vicinity of the QBO and fewer levels above the stratopause: 1000, 925, 850, 700, 600, 500, 400, 300, 250, 200, 170, 150, 130, 115, 100, 90, 80, 70, 60, 50, 40, 35, 30, 25, 20, 17, 15, 13, 11, 10, 9, 8, 6, 5, 4, 3, 2, 1.5, 1, 0.7, 0.5, and 0.4 hPa.

      In addition to the plev42 pressure-level output, Table B3 consist of quantities requested (i.e., hereafter "plevTEM") as these 6-hourly instantaneous fields will be used to calculate the TEM circulation offline in order to verify consistency with the

monthly TEM fields provided by the modeling centers in Table B2. Finally, the 6-hourly instantaneous output in Table B5 is also requested on the plevTEM vertical grid. To reduce data volumes, these fields may be provided in a 15°S–15°N latitude range, and only for the altitude range 150–0.4 hPa ($\sim$ 13–54 km), following the analogous request from QBOi. The lower altitude boundary of the requested range, 150 hPa ($\sim$ 13 km), is farther below the tropical tropopause than was requested in Phase 1 (Butchart et al., 2018, Table 4) so as to ensure adequate coverage of the near-tropopause region.

### 335  5.3   Temporal Resolution and Output Periods

For all years monthly mean output is requested for all variables and ensemble members. 6-hourly mean and daily mean data are requested for subsets of the diagnostics for the entire duration of the experiment, but for only one ensemble member. The

6-hourly (3-D) instantaneous output (Table B5) is also requested for one ensemble member, but for only 10 years (although 30 years is strongly encouraged).

## 5.4   Data Storage

As a working group within the QBOi APARC activity, QUOCA Phase 1 data will be uploaded and stored to the QBOi collective workspace on JASMIN, with eventual long-term archiving to the CEDA permanent archive. The current estimate of required data storage for 8 models contributing all Tier 1 and Tier 2 experiments is $\sim 40$ TB, and is based on the same byte-per-grid-cell value used in estimating storage for the CCMI Phase 2 experiments. Note that, while CMORizing of data is strongly encouraged, it is not required for hosting on the CEDA archive.

## 6   Discussion

There is growing evidence that simulating ozone feedbacks may be important for amplifying QBO temperatures in the lower stratosphere (Butchart et al. (2003); DallaSanta et al. (2021)) and mitigating the QBO's response to increased $CO_2$ (DallaSanta et al., 2021), although the robustness of these findings across models has not been assessed. While studies have exploited the CMIP6 archive to bin models into those with/without so-called "interactive chemistry", this approach does not control for other contributors to model structural uncertainty that can obscure the direct influence of stratospheric ozone on the circulation.

Building on the successes of previous phases of QBOi and CCMI, here we propose an experimental protocol in support of the new APARC cross-activity working group QUOCA that aims to improve understanding of ozone feedbacks on the QBO. Through its focus on questions **Q1-Q3**, this activity will complement the science priorities of Phases 2 of QBOi and CCMI, without introducing redundancies in either science scope or methodological approach. For example, whereas QBOi Phase 2 will employ a nudging framework (Hitchcock et al., 2022) to infer dynamical mechanisms underlying weaker-than-observed teleconnections in models, the QUOCA experiments will investigate the influence of ozone feedbacks on QBO teleconnections. At the same time, while the future scenario projections comprising Phase 2 of CCMI will primarily inform ozone recovery dates and the effects of geoengineering through stratospheric aerosol injection, the QUOCA future experiments will focus on understanding the driving mechanisms linking increased $CO_2$ concentrations to changes in the circulation and their modulation by ozone feedbacks.

While our primary focus is on the QBO, the QUOCA experiments can also be used to address broader questions related to ozone-circulation coupling (**Q4**). These issues – touching on ozone coupling with the BDC and polar vortices – might also benefit from an AMIP framework that isolates drivers of structural uncertainty distinct from global mean surface temperature, i.e., ozone changes, direct $CO_2$-induced radiative forcing (Calvo et al. (2025)).

In addition to contributing to enhanced understanding of ozone feedbacks, one practical deliverable from the QUOCA effort will be finer scrutiny of linearized ozone parameterizations which will be employed in some of the QUOCA models in lieu of more complex chemistry mechanisms (Table 3). Though not exhaustive enough a sample to make firm conclusions, this will present a rare opportunity to assess the degree to which the various ozone feedbacks quantified in both the present-day

and future experiments are captured by linearized parameterizations. Through its generation of new ozone-circulation coupling metrics, analysis of the QUOCA experiments will challenge parameterizations to reproduce more complex aspects of ozone-circulation coupling that may serve as new targets to guide model development.

Following community input received during two workshops held in November 2024 and March 2025, analysis of the QUOCA Phase 1 experiments will be organized around the following four themes:

– *Theme 1: Present-Day Ozone Feedback on Tropical Stratosphere: Circulation and Tracers*
   Analysis from this group will focus on understanding the present-day ozone feedback on the QBO and the QBO signature on the chemical and transport circulations.

  – *Theme 2: Ozone Feedback on Extratropical Circulation in Present-Day Climate*
   Analysis of this group will focus on understanding the present-day ozone feedback on QBO-polar vortex coupling and
the extratropical tropospheric circulation (e.g., midlatitude eddy-driven jets, North Atlantic Oscillation, etc.).

  – *Theme 3: Ozone Modeling: Analytical Models and Linearized Parameterizations*
   Analysis from this group will improve understanding of QBO-ozone coupling through analytical modeling (e.g., Randel et al. (2021); Ming et al. (2025)) and linearized ozone parameterizations.

  – *Theme 4: Ozone Feedback in a Warmer ($4xCO_2$) World: the BDC, the QBO, the Polar Vortex (and their Coupling)*
Analysis from this group will focus on understanding the ozone feedback on the $4xCO_2$ QBO response and its relationship to changes in the BDC, polar vortices and tropospheric circulation.

A key question not addressed by the QUOCA protocol is how well the models compare with observations. In particular, while comparisons of the simulated QBO and transport variations with observations will be straightforward, validations of the inferred ozone feedbacks will be challenging to perform without using models. One approach, however, might employ offline
radiative transfer calculations, which relate applied ozone perturbations to instantaneous heating rate changes (Randel et al., 2021), using both reanalysis and simulated fields as inputs as a means to bridge the gap between the observations and models. This work will likely fall under the domain of Working Group Theme 3 above.

Finally, certain aspects of the Phase 1 experimental design may limit immediate application of the results to certain problems. For example, by design, the FT experiments cannot be used to examine ozone feedbacks on climate sensitivity (Nowack et al.,
2015; Marsh et al., 2016). In addition, prescription of present-day ozone depleting substances (ODS) in the FT experiments assumes that the future ozone response will be dominated by transport ($CO_2$-driven) considerations. Both limitations may be addressed by repeating the FT experiments using either a coupled atmosphere-ocean framework and or future ODS scenarios in a Phase 2 protocol. However, more detailed development of such experiments will, of course, hinge on results produced by the working groups.

# Appendix A:  Forcings and Boundary Conditions

For all experiments, CMIP6-era forcings are strongly recommended and are available from the input4MIPs website[2]. There is some flexibility in specifying forcings, with the expectation that minor variations will not have an important effect on the experiment results, although it is essential that the same forcings are used for all experiments that are run by the same model. See Table 1 for more details.

**Ozone, Table 1 Col. 3**: For all interactive (INT) experiments, models should simulate interactive ozone, consistent with the transient SST and SIC boundary conditions and interactive chemistry of simulated ozone-depleting substances and other greenhouse gases, trace gases and tropospheric pollutants (e.g., $CH_4$, $N_2O$).

**Carbon Dioxide, Table 1 Col. 4**: Concentrations of $CO_2$ are recommended to follow forcings available from input4MIPs, so as to be consistent with SST and SIC climatological values over the 2000–2020 period.

**Other Trace Gas Concentrations, Table 1 Col. 5**: Concentrations of other radiatively active trace gases are recommended to follow forcings available from input4MIPs, so as to be consistent with SST and SIC climatological values over the 2000–2020 period.

**Ocean (SSTs and SICs), Table 1 Cols. 6 and 7**: AMIP boundary conditions for SSTs and SICs can be obtained from input4MIPs. Version `v20220201` of monthly `tosbcs` and `siconcbcs` from the PCMDI-AMIP 1.1.6 merged HadISST and NCEP OI2 product covers the required 2000–2020 time period needed to generate the climatological mean boundary condition.

**Volcanoes and Solar Cycle**: For consistency with the prescribed QBO and ocean boundary conditions, volcanic aerosol forcing averaged over the 2000–2020 period is recommended. Similarly, a prescribed 11-year solar cycle in solar total irradiance and UV irradiance is recommended.

# Appendix B:  Data Request

The QUOCA Phase 1 data request comprises 150 variables, as listed in Tables E1–E5. Of these, 68 were requested for the QBOi Phase 1 experiments. The additional 82 variables are newly requested and derive primarily from the CCMI Phase 2 data request (Table E4).

---

[2]https://esgf-node.llnl.gov/search/input4mips/

A few additional radiation and circulation diagnostics are also requested, the former for performing offline radiative heating kernel calculations as in Huang and Huang (2024). The additional 7 circulation variables will enable offline calculation of the Transformed-Eulerian Mean (TEM) (Table E3). These have been added to serve as a check on the TEM quantities provided by the modeling centers in Table E2, which should be calculated using 6-hourly instantaneous samples and then averaged to daily or monthly means, consistent with the method prescribed by DynVarMIP requirements for CMIP6 output (Gerber and Manzini, 2016, including their Corrigendum). That is, eddy covariances ($\overline{u'v'}$, etc.) should be computed by multiplying 6-hourly eddy quantities and then zonally averaging. All multiplicative products (i.e., multiplying eddy covariances by mean-flow terms to compute the EP flux components) should be computed at a 6-hourly frequency before time averaging to daily or monthly means. Furthermore, all calculations should be carried out on a pressure levels grid with vertical resolution comparable to the model resolution, so as to reduce errors in the computation of vertical derivatives (Gerber and Manzini, 2016). Final results can then be interpolated to the "plev42" standard pressure levels specified in Table E2.

## Appendix C: Prescribed Ozone in FT-NINT-4xCO$_2$ Experiment

Note that prescribing PD-NINT ozone concentrations in the FT-NINT-4xCO$_2$ integration will result in a disconnect between the 4xCO$_2$ (heightened) dynamical tropopause and the chemical tropopause implied in the prescribed ozone distribution. While this inconsistency can be corrected for either through prescription of the FT-INT-4xCO$_2$ ozone concentrations and/or ozone redistribution (for an example see Hardiman et al. (2019)), we clarify that the 4xCO$_2$ ozone feedback we seek to capture targets the following question: "How does the ozone response to 4xCO$_2$ (consisting of an ozone response to both a rise in tropopause height and an acceleration of the BDC) modulate the QBO-ozone feedback?" This question is distinct from asking how the ozone feedback captured by the PD-NINT and INT experiments (i.e., the mechanism initially proposed in Butchart et al. (2003)) changes under climate change. We privilege the former question, mainly because it is more relevant to CMIP6, in which most models used preindustrial control ozone concentrations in the 4xCO$_2$ experiment. The QUOCA Phase 1 experiments may therefore provide insight into circulation features in the CMIP6 ensemble that may be misrepresented as a result of ignoring ozone feedbacks on the climate's response to CO$_2$.

## Appendix D: Prescribing Stratospheric versus Tropospheric Ozone

The QUOCA project is focused primarily on improved understanding of the influence of interactive ozone dynamics *in the stratosphere*. Therefore, we strongly recommend that modeling centers with the ability to diagnostically distinguish between tropospheric and stratospheric ozone use this capability when performing the PD-NINT experiment. More precisely, for the PD-NINT experiment, we recommend that modeling centers prescribe the climatological ozone field from the PD-INT experiment only in the stratosphere, while maintaining interactive treatment of tropospheric ozone within the troposphere (identical to that employed in PD-INT). An obvious caveat with this approach is that different tropopause definitions among modeling centers will generate an additional degree of structural uncertainty, while also potentially resulting in unphysical results in regions

where the tropopause in PD-INT does not align with that in PD-NINT (see Tang et al. (2021) for more). Therefore, in cases where modeling centers adopt this approach we ask that centers provide information of which tropopause has been used and details about the methodology employed.

**Appendix E: Transient versus Time-Slice GISS E2-2 Sensitivity Results**

To test the sensitivity of our findings to methodological approach, we ran transient versions of the PD-NINT and PD-INT experiments using the GISS E2-2 model. Specifically, the PD-INT transient simulation is nearly identical to the $\sim 60$ year long (1960 - 2018) REF-D1 hindcast simulation requested as part of the CCMI Phase 2 effort supporting the 2022 WMO/UNEP Scientific Assessment of Ozone Depletion. PD-INT was run using full interactive chemistry and with forcing data (SSTs, SICs, compositional forcings (e.g., long-lived GHGs, ODS, etc.)) developed for CMIP6 and available through input4MIPs.

One important difference from the REF-D1 simulation, however, is that the QBO in PD-INT is generated internally in the model, whereas the REF-D1 protocol required that models nudge the QBO. The chemistry was spun up 10 years in this experiment.

After completing the PD-INT transient experiment we then performed a transient PD-NINT experiment, which is identical to PD-INT in terms of all forcings and boundary conditions, except that we prescribe a "QBO-filtered" (QBOf) three-dimensional

ozone field derived from the transient PD-INT experiment and keeping all other compositional and boundary forcings identical. This QBOf was constructed using a 3-year ozone running mean (i.e., January Year 2 smoothed ozone equals the average of January Year 1, January Year 2, and January Year 3) from the PD-INT experiment and then used to constrain a PD-NINT simulation. In addition, we also ran an intermediary step (referred to as "PD-NINT Transient Spec" in Figure E3) in which we prescribed the monthly three-dimensional ozone from PD-INT into the model to ensure that we could generate the same QBO

and circulation as in the PD-INT experiment through prescription (as opposed to online calculation of) the ozone fields.

Comparisons of Appendix Figures E1-E3 with Figures 3-5 show strikingly similar results between the transient and time-slice experiments, revealing little sensitivity of the main aspects of the ozone feedback on the QBO to methodology. In particular, for both time-slice (Figure 3) and transient (Figure E3) frameworks, interactive ozone leads to a reduced QBO period in the GISS E2-2 model by $\sim$ 2-3 months.

**Table E1.** Requested climate variable output. For variables with a vertical dimension, output is requested on the plev42 vertical grid.

| Name | Long name [Units] | Temporal Resolution | Spatial Dimensions |
|---|---|---|---|
| psl | Sea Level Pressure [Pa] | mon, day | longitude, latitude |
| ps | Surface Air Pressure [Pa] | mon, day | longitude, latitude |
| pr | Precipitation [kg m$^{-2}$ s$^{-1}$] | mon, day | longitude, latitude |
| prc | Convective Precipitation [kg m$^{-2}$ s$^{-1}$] | mon, day | longitude, latitude |
| rlut | Outgoing Longwave Radiation [W m$^{-2}$] | mon, day | longitude, latitude |
| rlds | Surface Downwelling Longwave Flux [W m$^{-2}$] | mon | longitude, latitude |
| rldscs | Clear-Sky Surface Downwelling Longwave Flux [W m$^{-2}$] | mon | longitude, latitude |
| rlus | Surface Upwelling Longwave Flux [W m$^{-2}$] | mon | longitude, latitude |
| rlutcs | Clear-Sky Outgoing Longwave Radiation [W m$^{-2}$] | mon | longitude, latitude |
| rsds | Surface Downwelling Shortwave Flux [W m$^{-2}$] | mon | longitude, latitude |
| rsdscs | Clear-Sky Surface Downwelling Shortwave Flux [W m$^{-2}$] | mon | longitude, latitude |
| rsdt | TOA Incident Shortwave Radiation[W m$^{-2}$] | mon | longitude, latitude |
| rsus | Surface Upwelling Shortwave Flux [W m$^{-2}$] | mon | longitude, latitude |
| rsuscs | Clear-Sky Surface Upwelling Shortwave Flux [W m$^{-2}$] | mon | longitude, latitude |
| rsut | TOA Shortwave Outgoing Flux [W m$^{-2}$] | mon | longitude, latitude |
| rsutcs | Clear-Sky TOA Shortwave Outgoing Flux [W m$^{-2}$] | mon | longitude, latitude |
| tas | Near-Surface Air Temperature [K] | mon, day | longitude, latitude |
| uas | Eastward Near-Surface Wind [m s$^{-1}$] | mon, day | longitude, latitude |
| vas | Northward Near-Surface Wind [m s$^{-1}$] | mon, day | longitude, latitude |
| ptp | Tropopause Air Pressure [Pa] | mon, day | longitude, latitude |
| tatp | Tropopause Air Temperature [K] | mon, day | longitude, latitude |
| ta | Air Temperature [K] | mon, day | longitude, latitude, plev42 |
| ua | Eastward Wind [m$^{-1}$] | mon, day | longitude, latitude, plev42 |
| va | Northward Wind [m$^{-1}$] | mon, day | longitude, latitude, plev42 |
| wap | Vertical Velocity, $\omega$(=dp/dt) [Pa$^{-1}$] | mon, day | longitude, latitude, plev42 |
| zg | Geopotential Height [m] | mon, day | longitude, latitude, plev42 |

**Table E2.** Requested radiative output and dynamical fields, most of which are zonally averaged. For variables with a vertical dimension, output is requested on the plev42 vertical grid.

| Name | Long name [Units] | Temporal Resolution | Spatial Dimensions |
|---|---|---|---|
| zmtnt | Total Temperature Tendency [K s$^{-1}$] | mon | latitude, plev42 |
| tntlwas | All sky longwave heating rate [K s$^{-1}$] | mon | latitude, plev42 |
| tntlwcs | Clear sky longwave heating rate [K s$^{-1}$] | mon | latitude, plev42 |
| tntswas | All short shortwave heating rate [K s$^{-1}$] | mon | latitude, plev42 |
| tntswcs | Clear sky shortwave heating rate [K s$^{-1}$] | mon | latitude, plev42 |
| ua | Eastward Wind [m s$^{-1}$] | mon | latitude, plev42 |
| va | Northward Wind [m s$^{-1}$] | mon | latitude, plev42 |
| wap | Vertical velocity, Omega(=dp/dt) [Pa s$^{-1}$] | mon | latitude, plev42 |
| ta | Air Temperature [K] | mon | latitude, plev42 |
| zg | Geopotential Height [m] | mon, day | latitude, plev42 |
| vtem | Transformed Eulerian Mean Northward Wind [m s$^{-1}$] | mon, day | latitude, plev42 |
| wtem | Transformed Eulerian Mean Upward Wind [m s$^{-1}$] | mon, day | latitude, plev42 |
| psitem | Transformed Eulerian Mean Mass Streamfunction [kg s$^{-1}$] | mon, day | latitude, plev42 |
| epfy | Northward Component of the Eliassen-Palm Flux [m$^3$ s$^{-2}$] | mon | latitude, plev42 |
| epfz | Upward Component of the Eliassen-Palm Flux [m$^3$ s$^{-2}$] | mon | latitude, plev42 |
| v'T' | Northward flux of temperature [m s$^{-1}$ K] | mon | latitude, plev42 |
| u'v' | Northward flux of eastward momentum [m$^2$ s$^{-2}$] | mon | latitude, plev42 |
| u'w' | Upward flux of eastward momentum [m$^2$ s$^{-2}$] | mon | latitude, plev42 |
| utendnet | Net tendency of eastward wind due to all parameterized processes [m s$^{-2}$] | mon, day | latitude, plev42 |
| utendogw | Tendency of eastward wind due to orographic gravity waves [m s$^{-2}$] | mon, day | latitude, plev42 |
| utendnogw | Tendency of eastward wind due to non-orographic gravity waves [m s$^{-2}$] | mon, day | latitude, plev42 |
| vtendnet | Net tendency of northward wind due to all parameterized processes [m s$^{-2}$] | mon, day | latitude, plev42 |
| vtendogw | Tendency of northward wind due to orographic gravity waves [m s$^{-2}$] | mon, day | latitude, plev42 |
| vtendnogw | Tendency of northward wind due to non-orographic gravity waves [m s$^{-2}$] | mon, day | latitude, plev42 |
| precip | Precipitation flux [kg m$^{-2}$ s$^{-1}$] | mon, day | latitude, longitude, plev42 |
| cod | Cloud optical depth [1] | mon, day | latitude, longitude |
| convec_cloud_area_frac | Convective cloud area [%] | mon, day | latitude, longitude |
| cloud_area_frac | Total cloud area [%] | mon, day | latitude, longitude |

**Table E3.** Requested 6-hourly zonally averaged instantaneous output needed to compute the TEM circulation offline as validation of the online TEM output in Table B2. Data is requested at a vertical resolution equivalent to the underlying model resolution (i.e., plevTEM).

| Name | Long name [Units] | Temporal Resolution | Spatial Dimensions |
|------|-------------------|---------------------|--------------------|
| ta | Air temperature [K] | 6-hourly | latitude, plevTEM |
| ua | Eastward wind [m s$^{-1}$] | 6-hourly | latitude, plevTEM |
| va | Northward wind [m s$^{-1}$] | 6-hourly | latitude, plevTEM |
| wap | Vertical velocity, Omega(=dp/dt) [Pa s$^{-1}$] | 6-hourly | latitude, plevTEM |
| v'T' | Northward flux of temperature [m s$^{-1}$ K] | 6-hourly | latitude, plevTEM |
| u'v' | Northward flux of eastward momentum [m$^2$ s$^{-2}$] | 6-hourly | latitude, plevTEM |
| u'w' | Upward flux of eastward momentum [m$^2$ s$^{-2}$] | 6-hourly | latitude, plevTEM |

**Table E4.** Requested chemical and idealized tracer output, most of which is zonally averaged. For variables with a vertical dimension, output is requested on the plev42 vertical grid. *Note these quantities will not make sense for tropospheric ozone, and if not diagnosed as such, need not be reported below the tropopause. Continued on the next page.

| Name | Long name [Units] | Temporal Resolution | Spatial Dimensions |
|---|---|---|---|
| $O_3$ | Ozone [ppm] | mon, day | latitude, plev42 |
| $H_2O$ | Water Vapor [ppm] | mon, day | latitude, plev42 |
| $O_3S$ | Stratospheric ozone tracer [ppm] | mon | latitude, plev42 |
| $O_3$STE | Net stratosphere-to-troposphere exchange O3 flux [Tg/year] | mon | latitude, longitude |
| AOD | Aerosol Optical Depth [1] | mon | latitude, longitude |
| e90 | Idealized uniform surface 90 day$^{-1}$ loss tracer [ppb] | mon | latitude, plev42 |
| AOA | Stratospheric mean age-of-air [years] | mon | latitude, plev42 |
| ST80$_{25}$ | Idealized stratospheric 25 day$^{-1}$ loss tracer [ppb] | mon | latitude, plev42 |
| $O_3$col | Total column ozone [ppm] | mon | latitude, longitude |
| NO | NO volume mixing ratio [ppb] | mon | latitude, plev42 |
| $NO_2$ | $NO_2$ volume mixing ratio [ppb] | mon | latitude, plev42 |
| $N_2O$ | $N_2O$ volume mixing ratio [ppb] | mon | latitude, plev42 |
| $N_2O_5$ | $N_2O_5$ volume mixing ratio [ppb] | mon | latitude, plev42 |
| $HNO_3$ | $HNO_3$ volume mixing ratio [ppb] | mon | latitude, plev42 |
| $NO_y$ | Total reactive nitrogen volume mixing ratio [ppb] | mon | latitude, plev42 |
| $Cl_2$ | Total inorganic chlorine volume mixing ratio [ppb] | mon | latitude, plev42 |
| $Br_2$ | Total inorganic bromine volume mixing ratio [ppb] | mon | latitude, plev42 |
| O | O volume mixing ratio [ppb] | mon | latitude, plev42 |

| Name | Long name [Units] | Temporal Resolution | Spatial Dimensions |
| --- | --- | --- | --- |
| Total Cl | Total Cl volume mixing ratio [ppb] | mon | latitude, plev42 |
| Total Br | Total Br volume mixing ratio [ppb] | mon | latitude, plev42 |
| ClO | ClO volume mixing ratio [ppb] | mon | latitude, plev42 |
| $ClNO_2$ | $ClNO_2$ volume mixing ratio [ppb] | mon | latitude, plev42 |
| $ClONO_2$ | $ClONO_2$ volume mixing ratio [ppb] | mon | latitude, plev42 |
| HCl | HCl volume mixing ratio [ppb] | mon | latitude, plev42 |
| BrO | BrO volume mixing ratio [ppb] | mon | latitude, plev42 |
| OH | OH volume mixing ratio [ppb] | mon | latitude, plev42 |
| $HO_2$ | $HO_2$ volume mixing ratio [ppb] | mon | latitude, plev42 |
| $CH_4$ | $CH_4$ volume mixing ratio [ppb] | mon | latitude, plev42 |
| $SO_2$ | $SO_2$ volume mixing ratio [ppb] | mon | latitude, plev42 |
| JO3 | $O_3$ photolysis frequency [$s^{-1}$] | mon | latitude, plev42 |
| JO2 | $O_2$ photolysis frequency [$s^{-1}$] | mon | latitude, plev42 |
| O3prod | $O_3$ production [ppbv $s^{-1}$] | mon | latitude, plev42 |
| O3loss | Chemical $O_3$ loss[ppbv $s^{-1}$] | mon | latitude, plev42 |
| O3lossOx* | Chemical $O_3$ loss by Ox [ppb $s^{-1}$] | mon | latitude, plev42 |
| O3lossHOx* | Chemical $O_3$ loss by HOx [ppb $s^{-1}$] | mon | latitude, plev42 |
| O3lossNOx* | Chemical $O_3$ loss by NOx [ppb $s^{-1}$] | mon | latitude, plev42 |
| O3lossClOx* | Chemical $O_3$ loss by ClOx [ppb $s^{-1}$] | mon | latitude, plev42 |
| SAD | Surface area density of sulfate aerosols [$m^{-1}$] | mon | latitude, plev42 |
| NAD | Surface area density of NAT PSC particles [$m^{-1}$] | mon | latitude, plev42 |
| PSC | Surface area density of water ice PSC particles [$m^{-1}$] | mon | latitude, plev42 |

**Table E5.** The 6-hourly instantaneous 3-D equatorial output for assessing equatorial wave spectra. Data is requested only for a subset of latitudes (15°S-15°N), pressure levels (vertical resolution equivalent to underlying model resolution between 150 hPa and 0.4 hPa), and 10 years of one ensemble member (although 30 years is strongly encouraged).

| Name | Long name [Units] | Temporal Resolution | Spatial Dimensions |
|------|-------------------|---------------------|--------------------|
| ta | Air Temperature [K] | 6-hourly | longitude, latitude, plevTEM |
| ua | Eastward Wind [m s$^{-1}$] | 6-hourly | longitude, latitude, plevTEM |
| va | Northward Wind [m s$^{-1}$] | 6-hourly | longitude, latitude, plevTEM |
| wap | Vertical velocity, Omega(=dp/dt) [Pa s$^{-1}$] | 6-hourly | longitude, latitude, plevTEM |

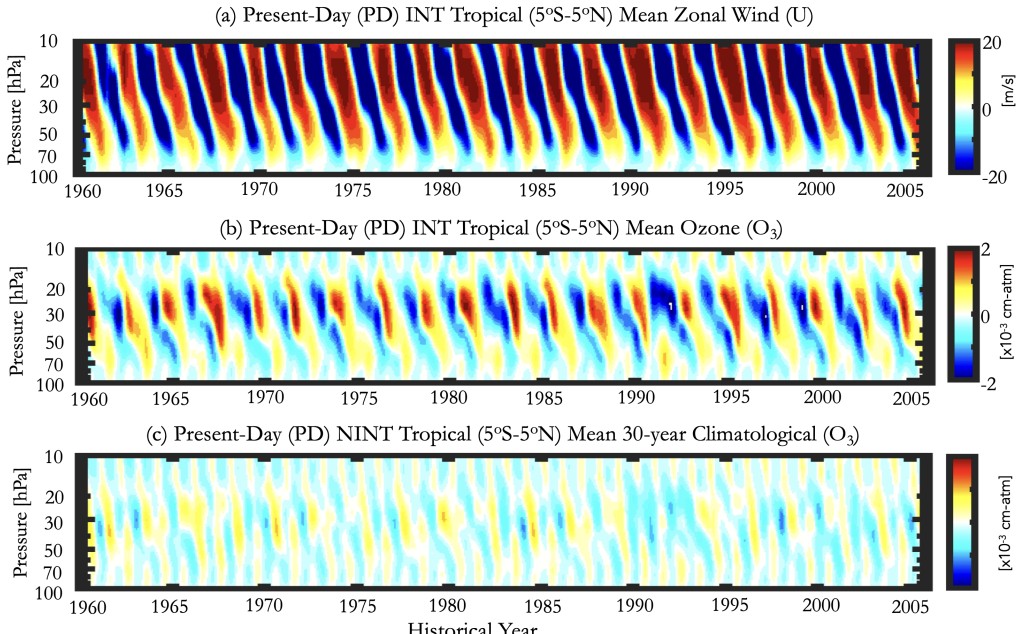

**Figure E1.** As in Figure 3, but for the transient integration.

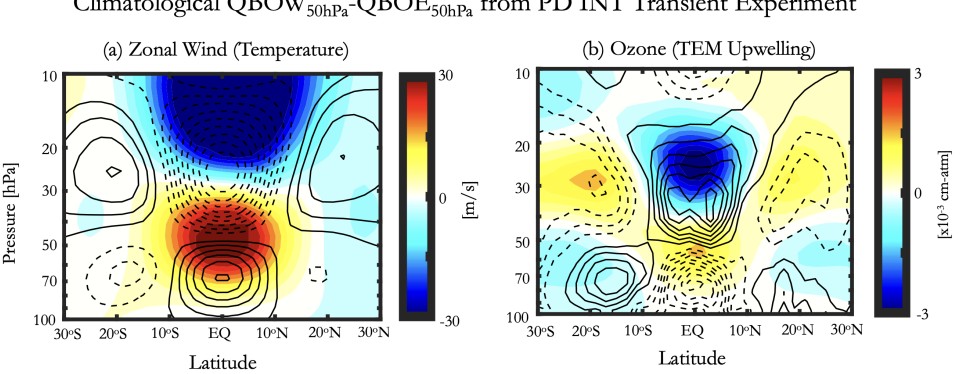

**Figure E2.** As in Figure 4, but for the transient integration.

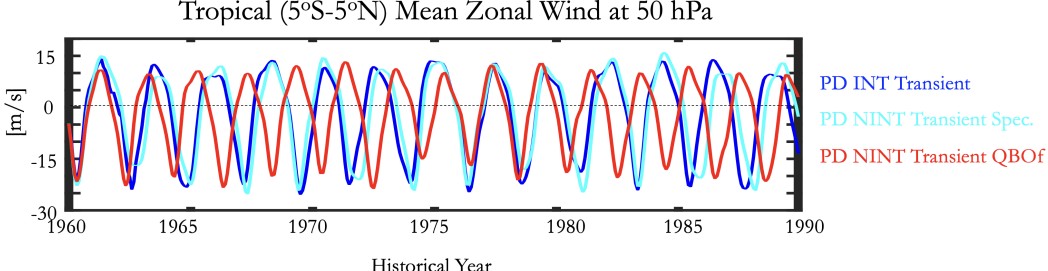

**Figure E3.** As in Figure 5, but for the transient integration.

*Code and data availability.* The input data and scripts used produce the plots for all the GISS E2-2 simulations presented in this paper are archived on repository under DOI:10.5281/zenodo.17063937 (Orbe, 2025). For information on the code availability for the individual models considered in this paper see the appropriate references given in Table 3. The data request is contained in a set of json files (CMOR tables) that give the attributes of the requested variables, and can be used in preparing the data with CMOR software as is done for CMIP. The QUOCA json files are available at https://github.com/QUOCA-project/cmor-tables.

*Author contributions.* The study was conceived by CO and AM. CO wrote the manuscript and performed the GISS simulations. CO, AM, GC and JA developed the data request. All authors contributed to editing of the manuscript.

*Competing interests.* The authors declare no competing interests.

*Acknowledgements.* We gratefully acknowledge support for the QUOCA working group from the Atmospheric Processes And their Role in Climate (APARC; https://www.aparc-climate.org/) core project of the World Climate Research Programme (WCRP; https://www.wcrp-climate. org/). We also gratefully acknowledge QBOi APARC activity and the UK Centre for Environmental Data Analysis (CEDA; https://www. ceda.ac.uk/) for hosting the QUOCA data archive and providing computing resources via the UK's collaborative data analysis environment JASMIN platform (https://jasmin.ac.uk/) (Lawrence et al., 2013). CO acknowledges climate modeling at GISS, which is supported by the NASA Modeling, Analysis, and Prediction program, and resources supporting this work were provided by the NASA High-End Computing (HEC) Program through the NASA Center for Climate Simulation (NCCS) at Goddard Space Flight Center. Part of the study was supported by the E3SM project at Lawrence Livermore National Laboratory (LLNL), funded by the U.S. Department of Energy, Office of Science, Office of Biological and Environmental Research through the Earth System Model Development program area. LLNL is operated by Lawrence Livermore National Security, LLC, for the U.S. Department of Energy, National Nuclear Security Administration under Contract DE-AC52-07NA27344. GC and AC acknowledge support from the European Union via the ERC Starting Grant, Number 101078127. The numerical simulations of CESM2-WACCM were performed using the NCAR Supercomputer Derecho. SW was supported by MEXT-Program for

the advanced studies of climate change projection (SENTAN) Grant Number JPMXD0722681344 and JSPS KAKENHI (JP22H01303 and JP23K22574). The numerical simulations of MIROC-ES2H were performed using the Earth Simulator of JAMSTEC.

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
