# Peer review of "Experimental Protocol for Phase 1 of the APARC QUOCA (QUasibiennial oscillation and Ozone Chemistry interactions in the Atmosphere) Working Group"

_EGUsphere, 2025_

## Referee Comment (RC1)

**Review of** "Experimental Protocol for Phase 1 of the APARC QUOCA (QUasibiennial oscillation and Ozone Chemistry interactions in the Atmosphere) Working Group"

by C. Orbe et al.

**Recommendation:** Minor revision

This is a useful summary of the experimental design for Phase 1 of the APARC QUOCA project, including a detailed set of tables that specify the model output requested. It is suitable for publication once the authors address the specific comments below.

**Specific Comments** (line number):

(66) "a dampening of": This should be "damping", as in "reducing", or "dissipating"—not as in "moistening" (dampening)

(71) "former claims": "previous claims" would be better.

(118) Figure 2, caption: "dampened" → damped

(141) "dampened" → damped

(146) "dampening" → damping

(150) "whether other models exhibit similar vertical structure": The version of WACCM used by Calvo et al. (JAS, 2015) does not appear to have the same response in the deep branch.

(179) "anomalous triggers (volcanoes, ENSO, wildfires)": However, among these, only ENSO is intrinsic to ocean-coupled simulations. Volcanoes and wildfires are boundary conditions that can be specified (or left out) as desired.

(184) "input4MIPs": This needs a link and/or reference.

(189) "at least ozone needs to run interactively": Note that the QBO-ozone perturbation above ~20 hPa will be driven mainly by changes in $NO_X$, which are due to advection. So, interactive $NO_X$ and its precursors ($N_2O$, tropospheric generation of $NO_X$ via lightning) should also be part of any INT experiment.

(192) "monthly annual cycle of three-dimensional ozone fields": This would be clearer as "annual cycle of three-dimensional, monthly-mean ozone fields".

(193) "The 90-year-long PD-INT experiment … three successive 30-year … climatologies": You may want to offer some motivation for using a separate 30-year climatology for each NINT experiment instead of driving all of them with a single 90-year climatology. I would not expect 30-year slices of the 90-year INT run to be significantly different. That is, the statistical

properties of 30 vs 90-year climatologies derived from the INT run ought to be statistically identical.

(209) "to ensure that the chemistry is sufficiently spun up": I do not understand this. The entire 90-year PD-INT run should be properly spun up because you are already asking for a 10-year spin-up before starting the 90-year PD-INT run. It seems to me that the real purpose of picking different times within the 90-day PD-INT run to initialize the FT-INT runs must be to ensure that each member of the PD-NINT ensemble is an independent realization. Am I misunderstanding this?

(230) "projects onto": "affects" would be clearer.

(238) "controlled more directly": Why "more directly"? The $O_X$ lifetime at and above 10 hPa is less than a month and becomes even shorter (1 day) by 0.5 hPa, so I think it is safe to say that $O_X$ (hence, ozone) is photochemically controlled at these altitudes.

(250) "the QBO period reduces": this should be "is reduced" or "decreases" (reduces is a transitive verb).

(262) "contributed submissions": "contributed results" or "submitted results" would be better.

(284) "tropopause air pressure and …": Will there be a common definition of "tropopause"? Will it be the same in the Tropics vs. extratropical latitudes?

(284) "Table B2": It would be useful to indicate in the table header that the quantities in the table are zonal averages.

(292) "subset of latitudes (15°S to 15°N)": ±20° might be more appropriate since forcing is substantial across this range of latitude (see, e.g., Garcia and Richter (2019), their figure 7).

(298) "including" → included

(306) "Table B4": Again, it would be useful to indicate in the table header that most of these are zonally averaged quantities, except for $O3_{STE}$, AOD and $ST80_{25}$. By the way, the table does not state what $ST80_{25}$ denotes; ditto for e90. This information should be included in the "long name" column, as is the case for other fields.

(312) "these will be used to calculate the TEM … to verify consistency … Table B2": I find this a bit confusing. I presume you do not intend to calculate TEM quantities from the eddy fluxes ($u'v'$, $v'T'$, etc.) in Table B2 (plev42 grid) but will instead compare TEM fields calculated from the outputs specified in Table B3 (plevTEM grid) with the *pre-calculate*d TEM quantities in Table B2 (vtem, wtem, psitem, epfy, epfz). Is that the idea? Note also that it is unlikely that the EP flux divergence, which involves the vertical derivative of epfz, can be calculated accurately from epfz output in Table B2 because the levels in that output are not native model levels. So, if one wants monthly div(F) on the plev42 grid, it ought to be included as a pre-computed field in table B2. I may be misunderstanding your intent, but I think this requires some clarification.

(316) "further below" → farther below

(321) "6-hourly (3-D) instantaneous output … for only 10 years": If this is going to be used to analyze QBO forcing by, say, compositing with respect to QBO phase, 10 years may not be enough for statistical reliability since that interval spans less than 5 QBO cycles.

---

## Author Comment (AC2)

**Response to Reviewer 1**

*Recommendation: Minor revision*

*This is a useful summary of the experimental design for Phase 1 of the APARC QUOCA project, including a detailed set of tables that specify the model output requested. It is suitable for publication once the authors address the specific comments below.*

➔ Thank you very much for your insightful and constructive review. Our responses to your specific comments are indicated below.

*Specific Comments (line number):*

*(66) "a dampening of": This should be "damping", as in "reducing", or "dissipating"—not as in "moistening" (dampening)*

➔ We thank the reviewer for catching this error. We have changed all incorrect references to"dampening" to "damping". Please see the revised manuscript.

*(71) "former claims": "previous claims" would be better.*

➔ We have changed "former claims" to "previous claims", as recommended by the reviewer. Please see the revised manuscript.

*(118) Figure 2, caption: "dampened" à damped*

➔ We thank the reviewer for spotting this error. This has been fixed in the revised manuscript.

*(141) "dampened" à damped*

➔ We thank the reviewer for spotting this error. This has been fixed in the revised manuscript.

*(146) "dampening" à damping*

➔ We thank the reviewer for spotting this error. This has been fixed in the revised manuscript.

*(150) "whether other models exhibit similar vertical structure": The version of WACCM used by Calvo et al. (JAS, 2015) does not appear to have the same response in the deep branch.*

➔ True – we thank the reviewer for her/his careful analysis. We have modified the sentence by adding "preliminary comparisons with other models suggest that the ozone feedback, particularly on the deep branch, may be model-dependent (Hufnagl et al. (2023), Calvo et al. (2025))". Please see the revised manuscript.

*(179) "anomalous triggers (volcanoes, ENSO, wildfires)": However, among these, only ENSO is intrinsic to ocean-coupled simulations. Volcanoes and wildfires are boundary conditions that can be specified (or left out) as desired.*

➔ We have retained the reference to volcanoes and wildfires, but we have changed the wording to more faithfully convey the difference between internal variability versus anomalous triggers. The rephrased clause reads as "internal variability intrinsic to atmosphere-ocean coupling (i.e., ENSO) and anomalous triggers (e.g., volcanoes, wildfires) in ozone." Please see the revised manuscript.

*(184) "input4MIPs": This needs a link and/or reference.*

➔ Thank you. We have added a link. Please see the revised manuscript.

*(189) "at least ozone needs to run interactively": Note that the QBO-ozone perturbation above ~20 hPa will be driven mainly by changes in NOX, which are due to advection. So, interactive NOX and its precursors (N2O, tropospheric generation of NOX via lightning) should also be part of any INT experiment.*

➔ We thank the reviewer for making this excellent point – indeed, several of the authors on this manuscript are interested in understanding precisely the NOx contribution to the QBO-ozone feedback at these levels. As shown in co-author Ming's recent study, NOx variations are essential, not only in reproducing the amplitude of the ozone peak above ~20 hPa, but it also has nonlocal effects on the ozone and temperature QBO patterns at lower altitudes (through modifying the column ozone aloft) (Ming et al. (2025)). At the same time, we intentionally do not ask that models use schemes with interactive NOx and its precursors as we want to include as many models as possible (including those with interactive QBOs, but linearized ozone mechanisms). One of the working groups in support of the QUOCA effort will be focused on assessing how well linearized ozone schemes capture various aspects of the ozone-QBO feedback (in particular, its vertical structure). Incorporating models with simplified ozone mechanisms (that do not have explicit NOx coupling) is therefore intentional. We now make these points explicit in the manuscript.

Ming, Alison, Peter Hitchcock, Clara Orbe, and Kimberlee Dubé. "Phase and amplitude relationships between ozone, temperature, and circulation in the quasi-biennial oscillation." *Journal of Geophysical Research: Atmospheres* 130, no. 4 (2025): e2024JD042469.

*(192) "monthly annual cycle of three-dimensional ozone fields": This would be clearer as "annual cycle of three-dimensional, monthly-mean ozone fields".*

➔ We agree with the reviewer and have changed the wording as suggested. Please see the revised manuscript.

*(193) "The 90-year-long PD-INT experiment … three successive 30-year … climatologies": You may want to offer some motivation for using a separate 30-year climatology for each NINT experiment instead of driving all of them with a single 90-year climatology. I would not expect 30-year slices of the 90-year INT run to be significantly different. That is, the statistical properties of 30 vs 90-year climatologies derived from the INT run ought to be statistically identical.*

➔ The reviewer makes a good point that one might not intuitively expect that the climatological ozone distributions generated from three distinct 30-year ozone periods would introduce as much intra-ensemble variability, compared to more standard practices for ensemble generation. However, preliminary experiments that we performed using the GISS E2-2 model (shown below) comparing this ensemble approach with a more conventional approach show that the resulting spread in the QBO is similar in magnitude. More precisely, the figure below shows that three NINT ensemble members (blue), generated using the proposed approach (i.e., each constrained with a distinct ozone climatology derived from non-overlapping 30-year windows from the PD INT integration) all show a reduced QBO period (by ~2.7-3.3 months) and reduced amplitude (by ~4.7-6.2 m/s).

Zonal Mean Zonal Wind (U), 5°S-5°N, 50 hPa (NINT (blue) vs. INT (red))

[Figure]

NINT perturbed ozone ensemble 1

$\omega$=24.8 months, $A$=24.4 m/s
$\omega$=27.5 months, $A$=30.0 m/s

NINT perturbed ozone ensemble 2

$\omega$=24.5 months, $A$=25.3 m/s

NINT perturbed ozone ensemble 3

$\omega$=24.2 months, $A$=23.8 m/s

Simulation Year

When we compare the intra-ensemble spread in this response to the responses in NINT simulations constrained with *same* ozone fields, but with slight differences in their initial atmospheric conditions (specifically, random perturbations in their tropospheric temperatures that are at most order ~1°C) we also find a reduction in QBO period with a similar spread (between ~2.8-3.1 months) and QBO amplitude (~4.9-6 m/s). This is shown below.

[Figure]

We conclude, therefore, that the ensemble spread generated through use of these different ozone fields is similar to that generated from more standard approaches. We have a brief description to the manuscript explaining our motivation for using this approach. Please see the revised manuscript.

*(209) "to ensure that the chemistry is sufficiently spun up": I do not understand this. The entire 90-year PD-INT run should be properly spun up because you are already asking for a 10-year spin-up before starting the 90-year PD-INT run. It seems to me that the real purpose of picking different times within the 90-day PD-INT run to initialize the FT-INT runs must be to ensure that each member of the PD-NINT ensemble is an independent realization. Am I misunderstanding this?*

➔ Wow—excellent catch! This was a complete error on our part, and we thank the reviewer for pointing out our mistake! We should have written "Each ensemble member is initialized from a different point in the PD-INT experiment in order to sample different initial states in the atmosphere." We are grateful to the reviewer's scrutiny of this section and refer her/him to the revised manuscript.

*(230) "projects onto": "affects" would be clearer.*

➔ Thanks – we have rephrased as suggested. Please see the revised manuscript.

*(238) "controlled more directly": Why "more directly"? The OX lifetime at and above 10 hPa is less than a month and becomes even shorter (1 day) by 0.5 hPa, so I think it is safe to say that OX (hence, ozone) is photochemically controlled at these altitudes.*

➔ OK – we have removed the reference to "more directly". Please see the revised manuscript.

*(250) "the QBO period reduces": this should be "is reduced" or "decreases" (reduces is a transitive verb).*

➔ Thanks – we have rephrased as suggested. Please see the revised manuscript.

*(262) "contributed submissions": "contributed results" or "submitted results" would be better.*

➔ OK – we revised this phrasing, changing it to "contributed results". Please see the revised manuscript.

*(284) "tropopause air pressure and …": Will there be a common definition of "tropopause"? Will it be the same in the Tropics vs. extratropical latitudes?*

➔ We have not asked that all modeling centers use the same definition of the tropopause and we realize that this will add another layer of structural uncertainty when interpreting the results, especially those related to stratosphere-troposphere exchange. However, we do ask in the appendix entitled "Prescribing Stratospheric versus Tropospheric Ozone" that modeling centers capable of diagnostically distinguishing between tropospheric and stratospheric ozone both a) prescribe the climatological ozone fields from the PD-INT experiment in the stratosphere only when performing the PD-NINT experiment; and b) provide information about which exact tropopause they have used and details of the methodology used to calculate it.

*(284) "Table B2": It would be useful to indicate in the table header that the quantities in the table are zonal averages.*

➔ Not all the quantities in the table are zonal averages (e.g., prec, cod, convec_cloud_area_frac, cloud_area_frac). We assume, however, that the reviewer (based on another comment) would like this noted, regardless. We have added a clause to the table caption qualifying that most of the variables are zonal averages. Please see the revised manuscript.

*(292) "subset of latitudes (15°S to 15°N)": ±20° might be more appropriate since forcing is substantial across this range of latitude (see, e.g., Garcia and Richter (2019), their figure 7).*

➔ While we understand the physical motivation for extending the latitude range as suggested, we want to maintain as much consistency with the QBOi output request as possible (as this may enable incorporation of results from those experiments as well in new analyses – mixing diagnostic definitions would obfuscate these types of efforts).

*(298) "including" à included*

➔ Thanks for spotting this error! We have fixed this typo in the revised manuscript.

*(306) "Table B4": Again, it would be useful to indicate in the table header that most of these are zonally averaged quantities, except for O3STE, AOD and ST8025. By the way, the table does not state what ST8025 denotes; ditto for e90. This information should be included in the "long name" column, as is the case for other fields.*

➔ OK – we have added the qualifier about zonal averages to the caption, as requested. We agree that more descriptions of the ST8025 and e90 were warranted. We have added these to the table, now referring to these tracers as stratospheric and surface loss tracers with 25 day$^{-1}$ and 80 day$^{-1}$ lifetimes, respectively. Please see the revised manuscript.

*(312) "these will be used to calculate the TEM … to verify consistency … Table B2": I find this a bit confusing. I presume you do not intend to calculate TEM quantities from the eddy fluxes (u'v', v'T', etc.) in Table B2 (plev42 grid) but will instead compare TEM fields calculated from the outputs specified in Table B3 (plevTEM grid) with the pre-calculated TEM quantities in Table B2 (vtem, wtem, psitem, epfy, epfz). Is that the idea? Note also that it is unlikely that the EP flux divergence, which involves the vertical derivative of epfz, can be calculated accurately from epfz output in Table B2 because the levels in that output are not native model levels. So, if one wants monthly div(F) on the plev42 grid, it ought to be included as a pre-computed field in table B2. I may be misunderstanding your intent, but I think this requires some clarification*

➔ Correct – we intend to calculate TEM quantities based on the eddy fluxes specified in Table B3 on the plevTEM grid. We are not exactly clear, however, why this is confusing to the reviewer because we already specify that the flux output from Table B3 (not B2) will be used. Nonetheless, we have added the qualifiers "6-hourly instantaneous" and "monthly" to more clearly distinguish between the B3 versus B2 outputs. Please see the revised manuscript.

*(316) "further below" à farther below*

➔ Thanks – this has been fixed. Please see the revised manuscript.

*(321) "6-hourly (3-D) instantaneous output … for only 10 years": If this is going to be used to analyze QBO forcing by, say, compositing with respect to QBO phase, 10 years may not be enough for statistical reliability since that interval spans less than 5 QBO cycles.*

➔ We agree that it may be challenging to extract a robust signature, but we are intentionally limiting this output request to ten years so that it remains tractable. No changes to the manuscript.

---

## Author Comment (AC3)

**Response to Reviewer 2**

*Summary*

*This paper is a data description article which introduces the APARC QUOCA. Personally, I planned to anticipate the initiation ceremony of this project, but I was hampered by some temporary assignments. I am happy to review this paper. Generally, this paper is well written and constructed, and this project is clearly introduced. The relationship between ozone and QBO is worth exploring. However, I also find that this study emphasizes the possible relationship between QBO and ozone from a dynamical perspective. Actually, for interactive ozone runs, the chemistry processes might also be responsible for the interaction between ozone and QBO. That is, the relative contribution of the ozone transport and chemical reaction for ozone change and therefore the QBO changes is not well shown. I suggest a revision at the present time (mostly minor points).*

**Major comments**

*1. The introduction mentions that the SST changes can more directly influence the QBO amplitude via modifying the Brewer Dobson circulation strength and the non-orographic gravity wave drag forcing. However, it is not easy to quantify the relative contribution of SST and the direct impact of ozone to the QBO amplitude. In my understanding, the SST might be the leading driver for the QBO amplitude and cycle changes. The interactive ozone limitedly affects the QBO amplitude, although some individual models indeed simulate a change of 10-20% in QBO amplitude.*

➔ We thank the reviewer for her/his thoughts. We agree that SSTs are the leading driver of the QBO amplitude and cycle changes, hence our incorporation of the "fixed SST" component of the future (FT) experiments, which have been designed to isolate this contribution. However, as we explain throughout, we are interested in the ozone feedback on the QBO – the reviewer asserts that this influence is small, but our point is that this influence has never been assessed across models in a rigorous apples-to-apples manner. The reviewer seems confident that ozone is a negligible feedback, but the available literature and our preliminary results do not seem to support that assertion.

*2. The largest obstacle for this project is to conquer the model bias in simulation of the QBO amplitude. First, models diverge in the simulate QBO amplitude, and the intermodel spread is much larger than the so-called improved change of the QBO amplitude with interactive ozone. Second, most CMIP5/6 models underestimate the QBO amplitude. Do you have any suggestion whereby the QBO amplitude bias and the so-called QBO amplitude change is well separated.*

➔ Of course – the intermodel spread in QBO amplitude and period is likely to be larger than the ozone feedback simulated within each model. We are not purporting to explain the intermodel spread in the QBO. Rather, we propose that ozone may be one of several – underlooked – contributors to QBO biases in models. The goal here really is to query if the sign of the ozone feedback – on QBO amplitudes, periods, secondary meridional circulation, etc. – is robust across models. Up to this point, a review of previous studies suggests that this is not the case. However, previous studies have also used widely diverging methodologies to define the ozone feedback and structure relevant experiments. To the best of our knowledge, our protocol represents the first attempt to assess the ozone feedback consistently across models, which we (and APARC leadership) assert is a worthy goal.

*3. This paper also plans to store the data on the admin server, where most of the community has no access to this dataset. I suggest to provide the data freely to all users, especially for beginners of the QBO works. CMIP6 data are available for global users. Since this project provides similar variables as in CMIP project, why not follow the CMIP6 project to share the data?*

➔ QUOCA originated as an extension of the APARC QBOi activity. Indeed, QUOCA was created partly as a response to APARC's directive to generate cross-activity working activities that leverage respective benefits from, in this case, QBOi (through QBO resolving models) and CCMI (through full chemistry models). In addition to the history of its nascence, we feel that APARC is, as stated in its mission statement, distinct from the CMIP enterprise in that it is primarily oriented at supporting process-based studies, not informing protocols. That is a critical reason why QUOCA best fits within APARC as its rather idealized experimental structure lends itself to asking questions that would otherwise be challenging to address using simulations constrained with more complex forcings.

*4. To take an example of using the APARC QUOCA project experiments, only one model provides the related runs. The GISS model indeed simulate the QBO amplitude change, but the consistency with other models is unknows. If one or two more models also provide the experiments and consistent projection at least with the same sign in change, the significance of this project is robustly confirmed. Figure 1 is a study from Butchart et al more than 20 years ago, which is too weak to support this study, and the model the authors used is also not mentioned. If they use a different model from GISS, please also show the relative change of the QBO amplitude with interactive ozone in this different model to support the results from GISS.*

➔ Shown below are preliminary results from the MIROC-E2H model. The other models' contributions are forthcoming. As the reviewer can see and per her/his comment "If one or two other models also provide experiment and consistent projection…" MIROC-ES2H shows an ozone feedback that is **consistent in sign** with what the GISS model shows. Specifically, the QBO period is longer and the QBO amplitudes are larger with ozone interactively calculated. The results from the GISS model, therefore, show promise in the direction toward robustness.

[Figure]

QUOCA PD Experiment Results from MIROC-ES2H

— INT
— NINT

Zonal Mean U, Averaged between 5°S-5°N (6-month running mean)

30 hPa

50 hPa

Year

*5. The GISS model the authors used is different from the CMIP6 version. I checked the GISS models in CMIP6, but the QBO is not simulated. Please provide relevant information for the difference between this model version and the CMIP6 model version in CMIP6. A general assessment of the model skill should be shown before the project is initialized. Low-skill models might not well answer the questions Q1-5 that is based on the models with a reasonable QBO simulation.*

➔ It is unfortunate that the reviewer is **not looking at the right version of the GISS model**. She/he is looking at E2.1 which, consistent with both its lower vertical resolution (40 levels) and lack of convective nonorographic gravity wave drag, does not have a QBO. Please, the reviewer needs to reformulate her/his comment by **looking at the \*\*E2.2\*\* submission.**

*6. The questions Q1-5 are repetitive in my understanding. Q1 is ozone => QBO and QBO teleconnections (including QBO downward impact to surface). Q2 is the identical to Q1 but for future scenarios. Ozone => QBO and QBO teleconnections (including large-scale circulation, BDC, and polar vortex). Q3 is repetitive for Q1 and Q2, but emphasizes the mechanism. What is the difference between mechanisms and dynamics? They are the same question. Further, the second question in Q3 is also biased from the main stream of the APARC QUOCA (comparison: CO2 => radiation, chemistry, and therefore circulation VS CO2 => SST changes => circulation). Q4 is 4xCO2 => BDC and polar vortices change => ozone feedback. Q5 is identical to Q4 but for the troposphere: 4xCO2 => troposphere => ozone feedback. Therefore, I suggest to focus on two or three key questions and do not list the question branches as the key questions.*

➔ The reason why Q3 is a separate question is because it requires not only the Tier 1, but Tier 2, experiments to be addressed (i.e., the experiments in which the 4xCO2 response is decomposed into contributions from increased SSTs vs. the direct CO2 radiative response). Since many modeling centers may only have the resources to contribute the Tier 1 experiments, it is practically useful to separate Q1-Q3 as we have done. The reviewer should understand that these are difficult times in the United States as computing (and overall

science) resources are heavily stressed, making it challenging for modeling groups to perform both Tier 1 and 2 runs. No changes to the manuscript regarding Q1-Q3.

That said, we do agree with the reviewer that Q4 and Q5 should be combined, particularly as both questions will be addressed by Working Group 4 (mentioned in the discussions). We thank the reviewer for highlighting this point. Please see the revised manuscript.

*Minor comments*

*1. L24: QBO can also impact the North Pacific pressure (geopotential height). Please refer to Rao et al. 2020a, 2020b (doi: 1175/JCLI-D-20-0024.1; doi: 10.1175/JCLI-D-19-0663.1).*

➔ Correct – we agree with the reviewer and have added these citations.

*2. L37: I am not sure if this feedback is seen in all CMIP6 models? Or this feedback is a result in very few models, and it is not extracted from observations.*

➔ Most CMIP6 models do not run with interactive ozone chemistry, limiting the ability to ask this question. Hence, the need for the QUOCA experimental protocol, so that we can ask this question using fit-for-purpose models and a clean framework.

*3. L44: This QBO biases should be well reviewed before this sentence. The weak QBO and the weak QBO teleconnections of both hemispheres have been reported in Rao et al. 2023ab (doi: ). You must provide the background how the QBO is simulated. The ozone can be prescribed, but the QBO can also be prescribed. If the interaction is focused, the experiment from prescribed ozone to free QBO and from prescribed QBO to free ozone can be setup.*

➔ The reviewer does not seem to appreciate the point of this paragraph, which is focused on identifying how the lack of interactive ozone may contribute (but not entirely explain) existing QBO biases. We understand that the main drivers of QBO biases are likely other aspects of model structural uncertainty, related to (lack of) vertical resolution (and implications for resolved wave forcing) and limitations of parameterizations of non-orographic gravity wave drag. However, these limitations have been well explored in previous studies, while the influence of ozone has not (the whole point of our manuscript). To satisfy the reviewer's concerns we have added one sentence highlighting the potential role of other sources of QBO biases, with appropriate citations, but have not provided a full in-depth review as it is not immediately relevant to the paragraph in question. Please note that the reviewer cites a Rao et al. (2023) study without a doi. Our search did not reveal any such relevant studies, so we have instead chosen to include the Richter et al. (2020) and Holt et al. (2022) references indicated below. We refer the reviewer to the revised manuscript.

Richter, Jadwiga H., James A. Anstey, Neal Butchart, Yoshio Kawatani, Gerald A. Meehl, Scott Osprey, and Isla R. Simpson. "Progress in simulating the quasi-biennial oscillation in CMIP models." *Journal of Geophysical Research: Atmospheres* 125, no. 8 (2020): e2019JD032362.

Holt, Laura A., François Lott, Rolando R. Garcia, George N. Kiladis, Yuan-Ming Cheng, James A. Anstey, Peter Braesicke et al. "An evaluation of tropical waves and wave forcing of the QBO in the QBOi models." *Quarterly Journal of the Royal Meteorological Society* 148, no. 744 (2022): 1541-1567.

*4. L48: Butchart et al. 2023 is frequently mentioned in this paper. The change in the QBO cycle is seen in this model, but inconsistent with other models (L53-54). So what can we learn from this inconsistency?*

➜ Does the reviewer mean Butchart et al. (2003)? Regarding the comment about consistency, that sentence refers to amplitude changes (not cycle/period changes). We ask that the reviewer clarify both the citation and the exact question that is being asked.

*5. L79: Projected QBO amplitude has been explored in Rao et al. 2020c GRL (doi: **1029/2020GL089149**). Please refer to this article for details. QBO is weakening, but the extratropical impact is strengthening.*

➜ First, line 79 does not refer to QBO amplitude. Please provide the correct line number. Second, we are not sure how the Rao et al. (2020c) reference is relevant to QUOCA as it has nothing to do with ozone (and its feedback on the QBO). Please clarify.

*6. L81-89: Some aspects of those three questions are repetitive.*

➜ We ask that the reviewer please clarify precisely which exact aspects are repetitive and in need of changing. It is challenging to make changes in response to a vague request/statement.

*7. L103: Can you show the results between 30-50 hPa where the QBO wind variability is largest?*

➜ Is the reviewer referring to Figure 1? These results are from a paper drafted more than twenty years ago (2003) and none of the authors on that study are on the list of authors on the present manuscript. That said, we can, of course, present to the reviewer the results from the GISS model. Is that of interest? Assuming "yes", here is the analogous figure to Figure 5, except now evaluated at 30 hPa (top) and 40 hPa (middle). We also include the original plot showing 50 hPa (bottom) so that the reviewer can compare all three levels easily. As she/he can see, the ozone feedback on QBO period (lengthen) and amplitude (increase) is consistent at all three (30,40,50 hPa) levels. We have added a sentence to the revised manuscript commenting on this overall consistency in the response at all three altitudes. We refer the reviewer to the revised manuscript.

[Figure]

*8. L104-105: Here you mention INT and NINT, which mean interactive and non-interactive ozone chemistry. In Figure 2, LINOZ is also an interactive chemistry, although it is simple.*

➔ Can the reviewer please clarify her/his point? What is the precise issue being raised here? Linearized ozone schemes are within the hierarchy of interactive chemistry schemes.

*9. L110: All ozone variability except the annual cycle is removed. It is not only the ozone related variation that has been removed, the variation related to SST forcing such as ENSO is also removed. This sentence is not accurate enough.*

➔ We have modified this sentence by adding "except the annual cycle". However, we ask that the reviewer please recall that these integrations are time-slice integrations, so ENSO is irrelevant here. Regardless, we have still made the requested change. Please see the revised manuscript.

*10. L111-112: Here it is a problem that should be addressed first. If the model is not consistent with observations, the dynamics processes analyzed from models are also unreliable. If the modeled annual cycle is biased from the observed annual cycle, what is the significance of the modelling results?*

➔ We privilege self-consistency over bias, meaning that models in which dynamics and chemistry are treated self-consistency can be used as tools to understand mechanisms. This assumption underlies the thousands of papers that have been written using CMIP and other modeling data to try and glean physical understanding. If the reviewer is suggesting that biased models cannot be used to understand mechanisms, then that would imply that no model-based study is instructive for anything, including the several papers that the reviewer asks that we cite. No changes to the manuscript.

*11. L116: The ozone feedback is the difference between interactive and non-interactive chemistry models. I suggest not to include the role of CO2, which mainly determines the mean background circulation and climate.*

➔ We appreciate the suggestion, but have decided to retain the experiments querying the ozone feedback in the 4xCO2 context. There are several pressing science questions related to the ozone feedback in a warmer world (not only related to the QBO, but also to the BDC, polar vortex, etc.) which have expanded the visibility and reach of QUOCA. No changes to the manuscript.

*12. Figure 2: The figure legend is misleading and hard to follow. In my understanding, OMA should be INT, to keep consistency with your description in the introduction section. LINOZ is also a type of INT, but you did not mention in the main part of the section 1.*

➔ We agree with the reviewer that the reference to "OMA" is unnecessarily confusing. We thank the reviewer for her/his suggestion and have replace "OMA" with "INT" in the figure legend. Please see the revised manuscript.

*13. Figure 2 caption: A typo here is picked out. FIXED is a type of non-interactive run, so it should be FT-NINT-1xCO2 (rather than FT-INT-1xCO2). Please clarify.*

➔ Excellent catch – we thank the reviewer for spotting this error! We have fixed this in the revised manuscript.

*14. L124: I agree with this statement. The SST is the determinant factor for the BDC change. Other improvement such as interactive ozone contribute very little to the total change of BDC.*

➔ There is no question that SSTs are the primary determinant of changes in the BDC in response to future increases in CO2. The reviewer makes a definitive statement about the impact of the ozone feedback on the BDC as though this was a well-established fact that had been reported in numerous studies. This is not the case and will be one of the science questions addressed by the QUOCA protocol.

*15. L128-129: I agree that the boundary conditions are of the first importance to induce the circulation change with global warming. The QBO changes might be very weak and is hard to extract.*

➔ We agree with the reviewer. No changes to the manuscript.

*16. L141: The dampened BDC response sounds like that the CO2 is more important than ozone for BDC response. Ozone only weakly dampens the total response. Ozone role in BDC change is indeed limited.*

➔ Again, we do not disagree with the reviewer that SSTs are the primary driver of the BDC response. No changes to the manuscript.

*17. L157-158: Very repetitive when compared with Q1-3. I have suggested to reconsider the key questions of this project.*

➔ We disagree that these are repetitive. Q1-Q3 are specifically concerned with the ozone feedback on the *QBO*. Q4 and Q5 (now merged into one question, at the recommendation of the reviewer) address other aspects of the large-scale stratospheric and tropospheric circulation.

*18. Figure 3c is repeating annual cycle of ozone without any QBO signals. What is the purpose for this experiment?*

➔ Figure 3 shows the ozone forcing that is used in the PD NINT experiment. It is important for transparency to explain/demonstrate clearly what we are doing in our experiments.

*19. L173-174 "while …": Here the CO2 response is mentioned. I disagree that the CO2 response exists. Since you have removed the role of global warming processed by using difference between two 4xCO2 runs. Here the 4xCO2 only creates a different mean state compared with the present-day climate. So it is none of the CO2 response business. That is, this sentence has entangled two questions. Please clarified.*

➔ The difference FT-INT-4xCO2 minus FT-NINT-4xCO2 represents the ozone feedback that occurs in the background context of 4xCO2. In other words, it captures the changes that occur **because ozone is responding to 4xCO2**. Similar definitions have been used in previous studies (Chiodo et al. (2018, 2019a,b)) and we are not inventing anything new here. Please refer to those studies. We have now referenced these studies in the manuscript to clarify this point. Please see the revised manuscript.

Chiodo, Gabriel, Lorenzo M. Polvani, Daniel R. Marsh, Andrea Stenke, W. Ball, Eugene Rozanov, Stefan Muthers, and Kostas Tsigaridis. "The response of the ozone layer to quadrupled CO 2 concentrations." *Journal of Climate* 31, no. 10 (2018): 3893-3907.

Chiodo, Gabriel, and Lorenzo M. Polvani. "The response of the ozone layer to quadrupled CO 2 concentrations: Implications for climate." *Journal of climate* 32, no. 22 (2019): 7629-7642.

*20. Table 1: FT-NINT-4xCO2, O3 should be "Climatological PD-INT" or "Same as above". I am not sure why the description is different from PD-NINT*

➔ Technically speaking, the description is correct as is, given that the PD-NINT climatology is derived from PD-INT. We agree with the reviewer, however, that this is probably unnecessary (not to mention there is an erroneous parenthesis that needs to be removed). This has been fixed in the revised manuscript.

*21. L184: I did not see input4MIPs in Table 1.*

➔ We have now added this to the table. Please see the revised manuscript.

*22. L195: I wonder if the three climatologies are different from each other. If so, how do you explain this difference?*

➔ The reviewer makes a good point that one might expect small differences among the climatologies and, therefore, small differences among the resulting ensemble members (constrained with these ozone fields). However, preliminary experiments that we performed

using the GISS E2-2 model (now shown below) comparing this ensemble approach with a more conventional approach show that the resulting spread in the QBO is similar in magnitude. More precisely, the figure below shows that three NINT ensemble members (blue), generated using the proposed approach (i.e., each constrained with a distinct ozone climatology derived from non-overlapping 30-year windows from the PD INT integration) all show a reduced QBO period (by ~2.7-3.3 months) and reduced amplitude (by ~4.7-6.2 m/s).

[Figure]

[Figure]

When we compare the intra-ensemble spread in this response to the responses in NINT simulations constrained with *same* ozone fields, but with slight differences in their initial atmospheric conditions (specifically, random perturbations in their tropospheric temperatures that are at most order ~1°C) we also find a reduction in QBO period with a similar spread (between ~2.8-3.1 months) and QBO amplitude (~4.9-6 m/s). This is shown below.

[Figure]

Zonal Mean Zonal Wind (U), 5°S-5°N, 50 hPa (NINT (blue) vs. INT (red))

NINT perturbed atmosphere ensemble 1
*ω*=24.7 months, *A*=25.1 m/s

NINT perturbed atmosphere ensemble 2
*ω*=24.4 months, *A*=24.0 m/s

We conclude, therefore, that the ensemble spread generated through use of these different ozone fields is similar to that generated from more standard approaches. We have added two sentences to the manuscript explaining our motivation for using this approach. Please see the revised manuscript.

*23. Once does not mean the improvement of those experiments. I still believe that the observational cycle shows some significance.*

➔ As discussed before, we privilege self-consistency within a given model (i.e., a self-consistent relationship between temperature, the circulation and ozone), even if that implies bias relative to observations. We refer to the reviewer to our response to the earlier related comment.

*24. L214-215: SST changes the property of QBO, which has been reported in Zhu and Rao 2025 (doi: /10.1016/j.atmosres.2025.108241)*

➔ Indeed, SST changes the QBO via direct and indirect (non-orographic gravity wave drag) means. This is well established and there is no need to include unecessary citations at this point.

*25. Figure 5: 30-yr and 50-yr PD NINT runs show nearly identical results. The cycle and amplitude of the QBO is nearly unchanged. What the mean QBO amplitude in the three runs if you use the method of Wang et al. 2025 (doi: 1016/B978-0-443-15638-0.00013-7)?*

➔ What motivation does the reviewer provide for raising this question? The Butchart et al. (2023) method applied here is quite standard and more widely applied than that of Wang et al. (2025). If the reviewer has physical grounding for suggesting that the Butchart et al. (2023) method is flawed, we ask that she/he provide this as motivation for performing new analysis. No changes to the manuscript.

*26. L228-229: The underestimation of the QBO amplitude should be included in the review of the existing literature in ample evidence (Anstey et al. 2019; Rao et al. 2020a, 2020b …)*

➔ We agree with the reviewer that more references should be included. We have included the suggested citations, at least the Rao, Garfinkel and White (2020) reference as this comments directly on the QBO representation in historical simulations (the other Rao et al. (2020) study is less relevant as it addresses future changes). Furthermore, it is not clear which Anstey et al. (2019) citation is being referenced, and we do not want to speculate on what the reviewer intended to write. Please see the revised manuscript.

*27. L234-236: Both models and observations provide evidence. Add some observation evidence. Lu et al. 2024 CAWE (doi: 1016/j.wace.2023.100627), 2025 Comm. (doi: 10.1038/s43247-024-01812-x)*

➔ We are more than happy to incorporate new citations. However, we are a bit concerned that the reviewer is asking us to cite studies that are not exactly relevant. In particular, these lines in the manuscript describe the process by which during the westerly phase of the QBO, anomalous downwelling associated with warmer anomalies with easterly wind shear bring air from the troposphere into the lower stratosphere, resulting in lower ozone. However, both of the references that the reviewer claims are critical to cite are not about the QBO. They are about (high latitude) stratospheric sudden warmings. Citing the proposed studies would be misleading at best, and, more likely, reflective of sloppy writing.

*28. L238-239: The role of the chemistry should be mentioned in the introduction earlier. Please tell readers clearly where dynamics dominates and where chemistry dominates.*

➔ We are confused by the reviewer's request here. She/he asks that we explain "clearly where dynamics dominates and where chemistry dominates". However, the lines that are being referenced literally state "While below 20 hPa, the ozone variations are more clearly modulated by transport associated with QBO dynamics, above 20 hPa, the ozone anomalies…[are] controlled more directly by variations in photochemistry." We provide the answer (~20 hPa) that the reviewer seeks. We ask that the reviewer please clarify what she/he is seeking.

*29. 245: As I point out in the major comments, why does the CMIP6 GISS model not simulate the QBO?*

➔ Again, the reviewer is looking at the **\*wrong model\***. We find it very concerning that the reviewer has not taken the time to ensure that she/he is looking at the right model. Please consider the correct model: **GISS E2-2**, not E2-1.

*30. L266: Figure 2 has shown the linear ozone chemistry, but the introduction to the linear chemistry appear here, much later than expected.*

➔ Our apologies, but there is no clear directive provided here. We will wait to hear back from the reviewer about when she/he expects that linearized ozone chemistry be introduced in the manuscript.

*31. Table 2, 3: I only have some knowledge that only four of those models can simulate the QBO (GEOSCCM, CESM2-WACCM, UKESM1, and MIROC6), if the model versions in CMIP5/6 is examined. Other models are unfamiliar to me and probably other readers.*

➔ We find it concerning that the reviewer is conflating her/his knowledge with the actual state of QBO simulations in models. In particular, while the reviewer may know of only these particular models, the additional models (unknown to the reviewer) are well-established leading chemistry models, many of which participated in CCMI. The reviewer appears to only know about the CMIP6 models but, of course, we are specifically leveraging the well-established chemistry climate modeling capabilities of the CCMI models to address this question. The reviewer does not seem to appreciate that we have been intentional in our attempt to bridge the resources from CCMI and QBOi, which is unfortunate. No changes to the manuscript.

32. *L320: data is => data are*

➔ We thank the reviewer for catching the error. This has been fixed in the revised manuscript.

33. *L325: JASMIN is only friendly to registered users. Do you have a plan B that share the datasets with more users?*

➔ As a working group within the QBOi activity our data archiving policy is consistent with what is being used in QBOi, which leverages JASMIN and CEDA. QBOi is considering the possibility of becoming a MIP within the CMIP7 enterprise, which would expand access to other users. However, QBOi (and QUOCA) are fundamentally process-oriented, not policy, multi-modeling exercises and therefore reside most naturally within APARC. Depending on how those discussions go there may be an eventual transition to CMIP7.

34. *L346: AMIP is good, but it lacks the air-sea interaction (or feedback from the sea). Can you have any better methods of assessing the role of the interactive ozone?*

➔ It is not clear exactly what the reviewer means by "better"? We already mentioned our concern with using a coupled atmosphere-ocean framework. As such, it is challenging to respond to this question without a clearer understanding of what the reviewer seeks. No changes to the manuscript.

35. *L358-369: Theme 1 + Theme 2 can be VS Theme 3.*

➔ We have already completed organization of the working groups and the most balanced (in size and expertise) situation consists of having contributors to (separate) Themes 1, 2 and 3. Not only is this scientifically defensible, it makes the most practical sense. No changes to the manuscript.

---

## Referee Report (RR1)

**Review of** "Experimental Protocol for Phase 1 of the APARC QUOCA (QUasibiennial oscillation and Ozone Chemistry interactions in the Atmosphere) Working Group" (revised)

by C. Orbe et al.

**Recommendation:** Accept after final (minor) revision

The authors have responded satisfactorily to most of my comments on the original paper. The revised version is suitable for publication after additional minor revision to address a few remaining specific comments listed below.

**Specific Comments** (line number):

(45) " inadequate vertical resolution":  Note that there is a new paper that examines systematically the impact of vertical resolution in NCAR's CESM model: Simpson et al., *J. Adv. Mod. Earth Sys.,* **17**, e2025MS004957, https://doi.org/10.1029/ 2025MS004957.

(183) "transient experiments may be complicated by": The only thing that unavoidably complicates analysis of coupled-ocean integrations is slowly evolving internal variability (e.g., ENSO). On the other hand, there is no reason why "anomalous triggers" (volcanoes, wildfires) need to be included in such integrations. At the risk of sounding pedantic, the point of the original comment (which the revision does not address) was that the "anomalous triggers" have nothing to do with why one would choose to carry out time-slice instead of transient runs.

(196) "the influence of NOx": This makes it sound like NOx is one among several factors that drive the ozone QBO above 20 hPa. However, NOx is the main catalytic loss mechanism for ozone between about 20 and 5 hPa (Brasseur and Solomon, 2005, Fig. 6.1), and several studies conclude that NOx mostly explains the ozone response in the upper stratosphere, e.g., Butchart et al. (*JGR* 2003), Anstey (*Nature Rev.* 2024), etc. I understand that the authors wish to have as many models as possible participate in the QUOCA exercise, but I wonder about the usefulness of any model that does not include a proper representation of NOx chemistry. However, since I am not too familiar with "simplified [ozone] mechanisms", I will defer to the authors in this matter.

(197) "nor at lower levels … column ozone aloft": It might be clearer to write something like "nor at lower levels, since the overlying ozone column can modulate the ultraviolet radiation that reaches the lower stratosphere and affect infrared transfer between layers".

(205) "similar in magnitude … more standard approaches": That is good to know but note that my original comment asked about the *motivation* for using separate 30-year segments to define the ozone climatology for the PD-NINT runs. I expected that 30-year segments would have statistical properties similar to the 90-year climatology (which you now show to be the case); however, I wondered whether there were specific reasons why your methodology would be preferable to the conventional approach. The answer to this question appears to be that there is no compelling reason to use 30-year segments but also no downside. (If I have understood this correctly, this comment does not require any additional revision).

(325) "Ming(2016))" → Ming (2016)

(325) "as close to … as possible": Why as close as possible to the native pressure grid? I would think one wants the output on the actual native grid to ensure accurate calculation of vertical derivatives. Or perhaps you are thinking about grid-cell midpoints vs. interfaces?

(327) "verify consistency with …Table B2": This is fine as long as such comparisons are limited to the TEM quantities included in Table B2. However, one quantity not included in that Table is the acceleration associated with the EP flux divergence due to resolved waves, div($\mathbf{F}$). Calculation of div($\mathbf{F}$) involves the vertical derivative of the vertical component of the EP flux, $F_z$, which cannot be calculated accurately from $F_z$ values interpolated to standard levels (plev42). On the other hand, all TEM quantities, including div($\mathbf{F}$), can be calculated offline from the output (Table B3) on the native model vertical grid (plevTEM).

---

## Author Response (AR2)

Dr. Clara Orbe
NASA Goddard Institute for Space Studies Code 611 New York · NY 10025
clara.orbe@nasa.gov
clara.orbe@gmail.com

Dr. Luke Western
Editor,
Geoscientific Model Development

September 24, 2025
re: manuscript number: egusphere-2025-2761 Title: "Experimental Protocol for Phase 1 of the APARC QUOCA (QUasibiennial oscillation and Ozone Chemistry interactions in the Atmosphere) Working Group"

Dear Dr. Western,

We thank the referees for their second round of reviews of our manuscript. We appreciate the care with which the reviewers have examined the manuscript throughout this process and have considered all feedback thoughtfully. In particular, we have addressed the additional 8 points raised by Reviewer 1, as well as the request to change the formatting of the appendix figures by Reviewer 2. Our responses to the two reviewers are addressed herein and reflected in the revised version of the manuscript.

With this revision we provide two versions of the revised manuscript, one of which includes the corrections highlighted in red. The point-by-point responses to the referees' comments are also included. We hope that the manuscript is now acceptable for publication in GMD. I confirm that my coauthors concur with the submission of our manuscript in its revised form. The revised version of the manuscript has been resubmitted electronically.

Yours sincerely,

Dr. Clara Orbe

**Response to Reviewer 1**

*Recommendation: Minor revision*

*(45) " inadequate vertical resolution": Note that there is a new paper that examines systematically the impact of vertical resolution in NCAR's CESM model: Simpson et al., J. Adv. Mod. Earth Sys., 17, e2025MS004957, https://doi.org/10.1029/ 2025MS004957.*

➔ Thank you for pointing us to this reference. We now cite this paper. Indeed, we were already familiar with it and find it interesting that vertical resolution does improve the QBO in that model, although this improvement does not translate to improved teleconnections. That certainly is an interesting result that warrants further research.

*(183) "transient experiments may be complicated by": The only thing that unavoidably complicates analysis of coupled-ocean integrations is slowly evolving internal variability (e.g., ENSO). On the other hand, there is no reason why "anomalous triggers" (volcanoes, wildfires) need to be included in such integrations. At the risk of sounding pedantic, the point of the original comment (which the revision does not address) was that the "anomalous triggers" have nothing to do with why one would choose to carry out time-slice instead of transient runs.*

➔ We maintain that "anomalous triggers" like volcanoes and wildfires can generate anomalies in stratospheric ozone that can be nontrivial to remove, as would be needed to generate the ozone annual cycle forcing fields used to constrain the PD NINT experiments. We do not agree with the reviewer that only ENSO – and not these types of events – would need to be considered. While this type of analysis would certainly not be impossible, it is likely that modeling centers would use a broad set of techniques to remove these influences from resulting ozone trends, generating even more (undesired) spread in the forcings among the models. No changes to the manuscript.

*(196) "the influence of NOx": This makes it sound like NOx is one among several factors that drive the ozone QBO above 20 hPa. However, NOx is the main catalytic loss mechanism for ozone between about 20 and 5 hPa (Brasseur and Solomon, 2005, Fig. 6.1), and several studies conclude that NOx mostly explains the ozone response in the upper stratosphere, e.g., Butchart et al. (JGR 2003), Anstey (Nature Rev. 2024), etc. I understand that the authors wish to have as many models as possible participate in the QUOCA exercise, but I wonder about the usefulness of any model that does not include a proper representation of NOx chemistry. However, since I am not too familiar with "simplified [ozone] mechanisms", I will defer to the authors in this matter.*

➔ We agree wholeheartedly with the reviewer that NOx will be very important. Indeed, we are eagerly anticipating results from two GISS model experiment submissions, which will compare the ozone feedback on the QBO generated from simplified versus fully interactive ozone schemes that respectively ignore and include NOx chemistry. Quantifying this effect within a single model will help isolate the contribution of NOx to the ozone QBO.

*(197) "nor at lower levels … column ozone aloft": It might be clearer to write something like "nor at lower levels, since the overlying ozone column can modulate the ultraviolet radiation that reaches the lower stratosphere and affect infrared transfer between layers".*

➔ OK—we have included this suggested rephrasing. While this does significantly lengthen the sentence, we agree that it makes the point clearer. Please see the revised manuscript.

*(205) "similar in magnitude … more standard approaches": That is good to know but note that my original comment asked about the motivation for using separate 30-year segments to define the ozone climatology for the PD-NINT runs. I expected that 30-year segments would have statistical properties similar to the 90-year climatology (which you now show to be the case); however, I wondered whether there were specific reasons why your methodology would be preferable to the conventional approach. The answer to this question appears to be that there is no compelling reason to use 30-year segments but also no downside. (If I have understood this correctly, this comment does not require any additional revision).*

➔ There is nothing special, per se, about 30 years versus more years. We just wanted to use enough years so that the statistical properties of each resulting climatology were not substantially different. The reviewer seems to appreciate this point. No changes to the manuscript.

*(325) "Ming(2016))" -> Ming (2016)*

➔ Thanks for catching this typo, which has been fixed in the revised version of the manuscript.

*(325) "as close to … as possible": Why as close as possible to the native pressure grid? I would think one wants the output on the actual native grid to ensure accurate calculation of vertical derivatives. Or perhaps you are thinking about grid-cell midpoints vs. interfaces?*

➔ Sorry, our original phrasing is unintentionally (and unnecessarily) confusing – we have now removed "as possible". Please see the revised manuscript.

*(327) "verify consistency with …Table B2": This is fine as long as such comparisons are limited to the TEM quantities included in Table B2. However, one quantity not included in that Table is the acceleration associated with the EP flux divergence due to resolved waves, div(F). Calculation of div(F) involves the vertical derivative of the vertical component of the EP flux, Fz, which cannot be calculated accurately from Fz values interpolated to standard levels (plev42). On the other hand, all TEM quantities, including div(F), can be calculated offline from the output (Table B3) on the native model vertical grid (plevTEM).*

➔ *Correct* -- comparisons are limited to the TEM quantities included in Table B2. No changes to the manuscript.

**Response to Reviewer 2**

*The revisions have largely improved the quality of the paper. I have no more questions. The only comment is that the figures in the appendix should be placed in portrait.*

➔ Thanks for taking the time to review our revised manuscript. As requested, we have changed the figures in the appendix to portrait mode.

---

## Author Response (AR3)

Dr. Clara Orbe
NASA Goddard Institute for Space Studies Code 611 New York · NY 10025
clara.orbe@nasa.gov
clara.orbe@gmail.com

Dr. Luke Western
Editor,
Geoscientific Model Development

October 1, 2025
re: manuscript number: egusphere-2025-2761 Title: "Experimental Protocol for Phase 1 of the APARC QUOCA (QUasibiennial oscillation and Ozone Chemistry interactions in the Atmosphere) Working Group"

Dear Dr. Western,

We thank the technical editor for her/his notes. On point 1: My co-author, Dr. Qi Tang (tang30@llnl.gov) confirms that the manuscript previously indicated as "Under Review" has now been accepted. The DOI is still (imminently) pending, however, so, should the technical editor need that information, she/he should contact Dr. Tang directly as I am currently furloughed and unable to complete official NASA duties. On point 2: The manuscript figures and tables all appear in the text before the references in the current version. On point 3: The figure in question has been modified to be color compliant.

I confirm that my coauthors concur with the submission of our manuscript in its revised form. The revised version of the manuscript has been resubmitted electronically.

Yours sincerely,

Dr. Clara Orbe